# RF-DETR: Neural Architecture Search for Real-Time Detection Transformers

**Isaac Robinson[1], Peter Robicheaux[1], Matvei Popov[1], Deva Ramanan[2], Neehar Peri[2]**
[1]Roboflow, [2]Carnegie Mellon University

## Abstract

Open-vocabulary detectors achieve impressive performance on COCO, but often fail to generalize to real-world datasets with out-of-distribution classes not typically found in their pre-training. Rather than simply fine-tuning a heavy-weight vision-language model (VLM) for new domains, we introduce RF-DETR, a light-weight specialist detection transformer that discovers accuracy-latency Pareto curves for any target dataset with weight-sharing neural architecture search (NAS). Our approach fine-tunes a pre-trained base network on a target dataset and evaluates thousands of network configurations with different accuracy-latency tradeoffs *without re-training*. Further, we revisit the "tunable knobs" for NAS to improve the transferability of DETRs to diverse target domains. Notably, RF-DETR significantly improves over prior state-of-the-art real-time methods on COCO and Roboflow100-VL. RF-DETR (nano) achieves 48.0 AP on COCO, beating D-FINE (nano) by 5.3 AP at similar latency, and RF-DETR (2x-large) outperforms GroundingDINO (tiny) by 1.2 AP on Roboflow100-VL while running $20\times$ as fast. To the best of our knowledge, RF-DETR (2x-large) is the first real-time detector to surpass 60 AP on COCO. Our code is available on GitHub.

## 1 Introduction

Object detection is a fundamental problem in computer vision that has matured in recent years (Felzenszwalb et al., 2009; Lin et al., 2014; Ren et al., 2015). Open-vocabulary detectors like GroundingDINO (Liu et al., 2023) and YOLO-World (Cheng et al., 2024) achieve remarkable zero-shot performance on common categories like `car`, `truck`, and `pedestrian`. However, state-of-the-art vision-language models (VLMs) still struggle to generalize to out-of-distribution classes, tasks and imaging modalities not typically found in their pre-training (Robicheaux et al., 2025). Fine-tuning VLMs on a target dataset significantly improves in-domain performance at the cost of runtime efficiency (due to heavy-weight text encoders) and open-vocabulary generalization. In contrast, specialist (i.e., closed-vocabulary) object detectors like D-FINE (Peng et al., 2024) and RT-DETR (Zhao et al., 2024) achieve real-time inference, but underperform fined-tuned VLMs like GroundingDINO. In this paper, we modernize specialist detectors by combining internet-scale pre-training with real-time architectures to achieve state-of-the-art performance *and* fast inference.

**Are Specialist Detectors Over-Optimized for COCO?** Sustained progress in object detection can be largely attributed to standardized benchmarks like PASCAL VOC (Everingham et al., 2015) and COCO (Lin et al., 2014). However, we find that recent specialist detectors implicitly overfit to COCO at the cost of real-world performance using bespoke model architectures, learning rate schedulers, and augmentation schedulers. Notably, state-of-the-art object detectors like YOLOv8 (Jocher et al., 2023) generalize poorly to real-world datasets with significantly different data distributions from COCO (e.g., number of objects per image, number of classes, and dataset size). To address these limitations, we present RF-DETR, a scheduler-free approach that leverages internet-scale pre-training to generalize to real-world data distributions. To better specialize our model for diverse hardware platforms and dataset characteristics, we revisit neural architecture search (NAS) in the context of end-to-end object detection and segmentation.

**Rethinking Neural Architecture Search (NAS) for DETRs.** NAS discovers accuracy-latency tradeoffs by exploring architectural variants within a pre-defined search space. NAS has been previously studied in the context of image classification (Tan & Le, 2019; Cai et al., 2019) and for model

sub-components like detector backbones Tan et al. (2020) and FPNs Ghiasi et al. (2019). Unlike prior work, we explore *end-to-end* weight-sharing NAS for object detection and segmentation. Our key insight, inspired by OFA (Cai et al., 2019), is that we can vary model inputs like image resolution, and architectural components like patch size during training. Further, weight-sharing NAS allows us to modify inference configurations like the number of decoder layers and query tokens to specialize our strong base model *without fine-tuning*. We evaluate all model configurations with grid search on a validation set. Importantly, our approach does not evaluate the search space until the base model has been fully-trained on the target dataset. As a result, all possible sub-nets (i.e., model configurations within the search space) achieve strong performance without further fine-tuning, significantly reducing the computational cost of optimizing for new hardware. Interestingly, we find that sub-nets not explicitly seen during training still achieve high performance (Appendix **??**), suggesting that RF-DETR can generalize to unseen architectures. Extending RF-DETR for segmentation is also relatively straightforward and only requires adding a lightweight instance segmentation head. We denote this model as RF-DETR-Seg. Notably, this allows us to also leverage end-to-end weight-sharing NAS to discover Pareto optimal architectures for real-time instance segmentation.

**Standardizing Latency Evaluation.** We evaluate our approach on COCO (Lin et al., 2014) and Roboflow100-VL (RF100-VL) (Robicheaux et al., 2025) and achieve state-of-the-art performance among real-time detectors. RF-DETR (nano) outperforms D-FINE (nano) by 5% AP on COCO at comparable run-times, and RF-DETR (2x-large) beats GroundingDINO (tiny) on RF100-VL at a fraction of the runtime. RF-DETR-Seg (nano) outperforms YOLOv11-Seg (x-large) on COCO while running $4 \times$ as fast. However, comparing RF-DETR's latency with prior work remains challenging because reported latency evaluation varies significantly between papers. Notably, each new model re-benchmarks the latency of prior work for fair comparison on their hardware. For example, D-FINE's reported latency evaluation of LW-DETR (Chen et al., 2024a) is 25% faster than originally reported. We identify that this lack of reproducibility can be primarily attributed to GPU power throttling during inference. We find that buffering between forward passes limits power over-draw and standardizes latency evaluation (Table 1).

**Contributions.** We present three major contributions. First, we introduce RF-DETR, a family of scheduler-free NAS-based detection and segmentation models that outperform prior state-of-the-art on RF100-VL (Robicheaux et al., 2025) and real-time methods with latencies $\leq 40$ ms on COCO (Lin et al., 2014) (Figure 1). To the best of our knowledge, RF-DETR is the first real-time detector to exceed 60 mAP on COCO. Next, we explore the "tunable-knobs" for weight-sharing NAS to improve accuracy-latency tradeoffs for end-to-end object detection (Figure 3). Notably, our use of a weight-sharing NAS allows us to leverage large-scale pre-training and effectively transfer to small datasets (Table 4). Lastly, we revisit current benchmarking protocols for measuring latency and propose a simple standardized procedure to improve reproducibility.

## 2 RELATED WORKS

**Neural Architecture Search (NAS)** automatically identifies families of model architectures with different accuracy-latency tradeoffs (Zoph & Le, 2016; Zoph et al., 2018; Real et al., 2019; Cai et al., 2018a). Early NAS approaches (Zoph & Le, 2016; Real et al., 2019) focused primarily on maximizing accuracy, with little consideration for efficiency. As a result, discovered architectures (e.g., NASNet and AmoebaNet) were often computationally expensive. More recent hardware-aware NAS methods (Cai et al., 2018b; Tan et al., 2019; Wu et al., 2019) address this limitation by incorporating hardware feedback directly into the search process. However, these methods must repeat the search and training process for each new hardware platform. In contrast, OFA (Cai et al., 2019) proposes a weight-sharing NAS that decouples training and search by simultaneously optimizing thousands of sub-nets with different accuracy-latency tradeoffs. Contemporary methods typically evaluate NAS for object detection by simply replacing standard backbones with NAS backbones in existing detection frameworks. Unlike prior work, we directly optimize end-to-end object detection accuracy to find Pareto optimal accuracy-latency tradeoffs for any target dataset.

**Real-Time Object Detectors** are of significant interest for safety-critical and interactive applications. Historically, two-stage detectors like Mask-RCNN (He et al., 2017) and Hybrid Task Cascade (Chen et al., 2019) achieved state-of-the-art performance at the cost of latency, while single-stage detectors like YOLO (Redmon et al., 2016) and SSD (Liu et al., 2016) traded accuracy for state-

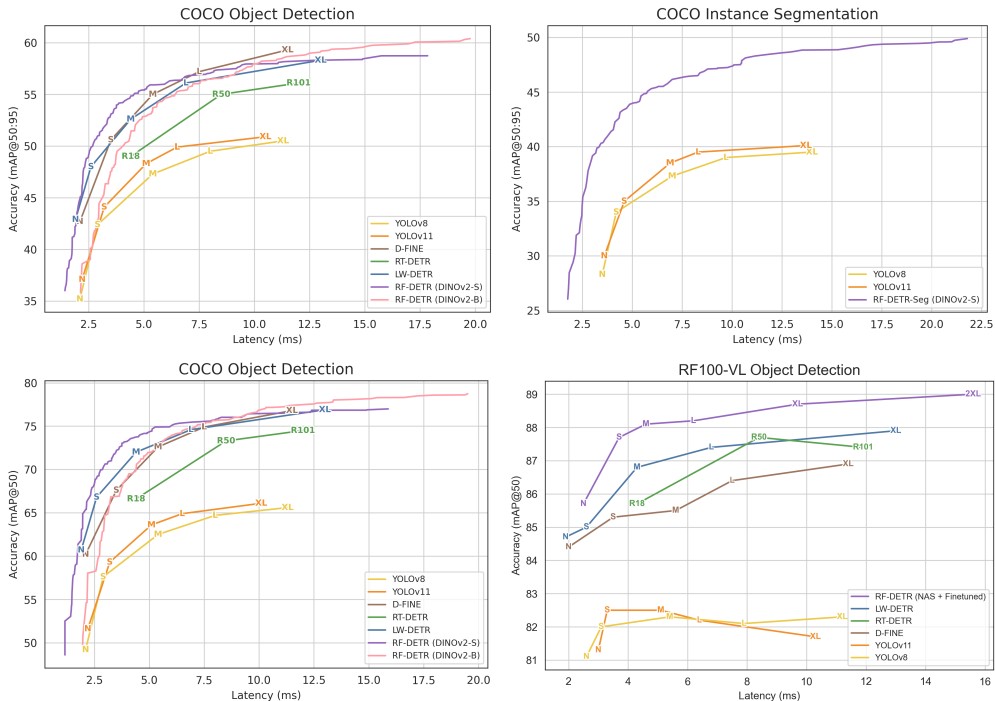

Figure 1: **Accuracy-Latency Pareto Curve.** We plot the Pareto accuracy-latency frontier for real-time detectors on the COCO detection val-set (top left, bottom left), COCO segmentation val-set (top right), and RF100-VL test-set (bottom right). Since RF100-VL contains 100 distinct datasets, we select target latencies for the N, S, M, L, XL, 2XL configurations, search for RF-DETR models with latencies within 10% of the target and report their average performance after fine-tuning to convergence. Importantly, all points along RF-DETR's continuous Pareto curves for COCO are derived from a single training run.

of-the-art runtime. However, modern detectors (Zhao et al., 2024) reexamine this accuracy-latency tradeoff, simultaneously improving on both axes. Recent YOLO variants innovate on architecture, data augmentation, and training techniques (Redmon et al., 2016; Wang et al., 2023; 2024; Jocher et al., 2023; 2024) to improve performance while maintaining fast inference. Despite their efficiency, most YOLO models rely on non-maximum suppression (NMS), which introduces additional latency. In contrast, DETR (Carion et al., 2020) removes hand-crafted components like NMS and anchor boxes. However, early DETR variants (Zhu et al., 2020; Zhang et al., 2022a; Meng et al., 2021; Liu et al., 2022) achieved strong accuracy at the cost of runtime, limiting their use in real-time applications. Recent works such as RT-DETR (Zhao et al., 2024) and LW-DETR (Chen et al., 2024a) have successfully adapted high performance DETRs for real-time applications. Building on LW-DETR, RF-DETR is the first real-time detector to achieve more than 60 AP on COCO.

**Vision-Language Models** are trained on large-scale, weakly supervised image-text pairs from the web. Such internet-scale pre-training is a key enabler for open-vocabulary object detection (Liu et al., 2023; Cheng et al., 2024). GLIP (Li et al., 2022) frames detection as phrase grounding with a single text query, while Detic (Zhou et al., 2022) boosts long-tail detection using ImageNet-level supervision (Russakovsky et al., 2015). MQ-Det (Xu et al., 2024) extends GLIP with a learnable module that enables multi-modal prompting. Recent VLMs demonstrate strong zero-shot performance and are often applied as black-box models in diverse downstream tasks (Ma et al., 2023; Peri et al., 2023; Khurana et al., 2024; Osep et al., 2024; Takmaz et al., 2025). However, Robicheaux et al. (2025) find that such models perform poorly when evaluated on categories not typically found in their pre-training, requiring further fine-tuning. In addition, many vision-language models are prohibitively slow, making them difficult to use for real-time tasks. In contrast, RF-DETR combines the fast inference of real-time detectors with the internet-scale priors of VLMs to achieve state-of-the-art performance on RF100-VL and at all latencies $\leq 40$ ms on COCO.

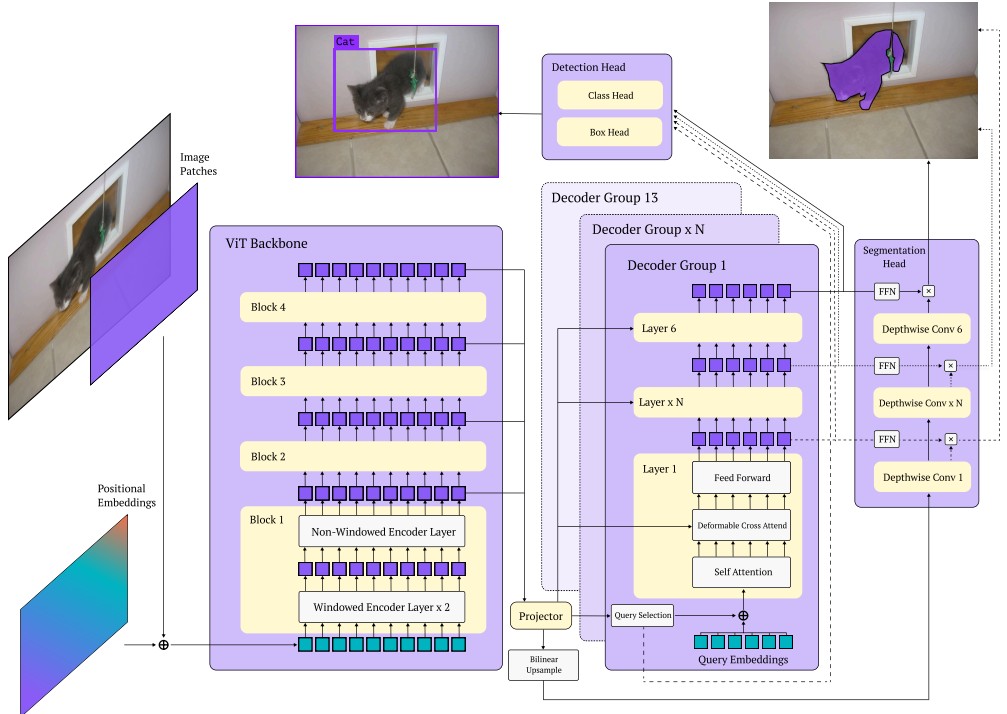

Figure 2: **RF-DETR Architecture**. RF-DETR uses a pre-trained ViT backbone to extract multi-scale features of the input image. We interleave windowed and non-windowed attention blocks to balance accuracy and latency. Notably, the deformable cross-attention layer and segmentation head both bilinearly interpolate the output of the projector, allowing for consistent spatial organization of features. Lastly, we apply detection and segmentation losses at all decoder layers to facilitate decoder drop out at inference.

# 3 RF-DETR: WEIGHT-SHARING NAS WITH FOUNDATION MODELS

In this section, we describe the architecture of our base model (Figure 2) and present the "tunable knobs" of our weight-sharing NAS (Figure 3). Further, we highlight the limitations of hand-designed learning-rate and augmentation schedulers, and advocate for a scheduler-free approach.

**Incorporating Internet-Scale Priors.** RF-DETR modernizes LW-DETR (Chen et al., 2024a) by simplifying its architecture and training procedure to improve generalization to diverse target domains. First, we replace LW-DETR's CAEv2 (Zhang et al., 2022b) backbone with DINOv2 (Oquab et al., 2023). We find that initializing our backbone with DINOv2's pre-trained weights significantly improves detection accuracy on small datasets. Notably, CAEv2's encoder has 10 layers with a patch size of 16, while DINOv2's encoder has 12 layers. Our DINOv2 backbone has more layers and is slower than CAEv2, but we make up for this latency using NAS (discussed next). Lastly, we facilitate training on consumer-grade GPUs via gradient accumulation by using layer norm instead of batch norm in the multi-scale projector.

**Real-Time Instance Segmentation.** Inspired by Li et al. (2023), we add a lightweight instance segmentation head to jointly predict high quality segmentation masks. Our segmentation head bilinearly interpolates the output of the encoder and learns a lightweight projector to generate a pixel embedding map. Specifically, we upsample the same low-resolution feature map for the detection and segmentation heads to ensure that it contains relevant spatial information. Unlike MaskDINO (Li et al., 2023), we do not incorporate multi-scale backbone features in our segmentation head to minimize latency. Lastly, we compute the dot product of all projected query token embeddings (at the output of each decoder layer transformed by a FFN) with the pixel embedding map to generate segmentation masks. Interestingly, we can interpret these pixel embeddings as segmentation

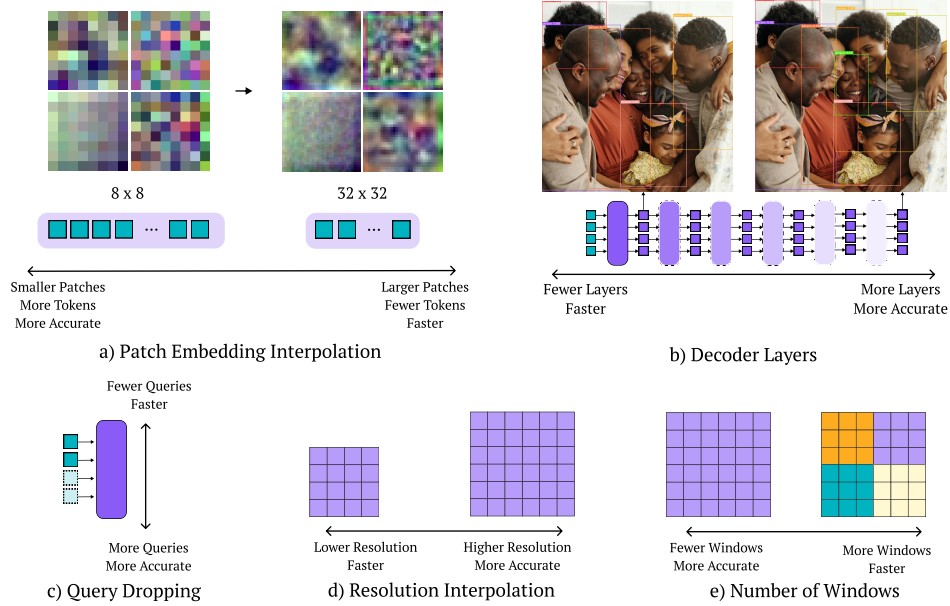

Figure 3: **NAS Search Space**. We vary (a) patch size, (b) number of decoder layers, (c) number of queries, (d) image resolution, and (e) number of windows per attention block when evaluating different operating points along RF-DETR's Pareto curve. In addition to training thousands of network configurations in parallel, we find that this "architecture augmentation" serves as a regularizer and improves generalization.

prototypes (Bolya et al., 2019). Motivated by LW-DETR's observation that pre-training improves DETRs, we pre-train RF-DETR-Seg on Objects-365 (Shao et al., 2019) psuedo-labeled with SAM2 (Ravi et al., 2024) instance masks.

**End-to-End Neural Architecture Search.** Our weight-sharing NAS evaluates thousands of model configurations with different input image resolutions, patch sizes, window attention blocks, decoder layers, and query tokens. At every training iteration, we uniformly sample a random model configuration and perform a gradient update (Appendix **??**). This allows our model to efficiently train thousands of sub-nets in parallel, similar to ensemble learning with dropout (Srivastava et al., 2014). We find that this weight-sharing NAS approach also serves as a regularizer during training, effectively performing "architecture augmentation". To the best of our knowledge, RF-DETR is the first end-to-end weight-sharing NAS applied to object detection and segmentation. We describe each component below.

- *Patch Size*. Smaller patches lead to higher accuracy at greater computational cost. We adopt a FlexiViT-style (Beyer et al., 2023) transformation to interpolate between patch sizes during training.

- *Number of Decoder Layers*. Similar to recent DETRs (Peng et al., 2024; Zhao et al., 2024), we apply a regression loss to the output of all decoder layers during training. Therefore, we can drop any (or all) decoder blocks during inference. Interestingly, removing the entire decoder during inference effectively turns RF-DETR into a single-stage detector. Notably, truncating the decoder also shrinks the size of the segmentation branch, allowing for greater control over segmentation latency.

- *Number of Query Tokens*. Query tokens learn spatial priors for bounding box regression and segmentation. We drop query tokens (ordered by the maximum sigmoid of the corresponding class logit per token at the output of the encoder, see Appendix **??**) at test time to vary the maximum number of detections and reduce inference latency. The Pareto optimal number of query tokens implicitly encodes dataset statistics about the average number of objects per image in a target dataset.

- *Image Resolution.* Higher resolution improves small object detection performance, while lower resolution improves runtime. We pre-allocate $N$ positional embeddings corresponding to the largest image resolution divided by the smallest patch size and interpolate these embeddings for smaller resolutions or larger patch sizes.

- *Number of Windows per Windowed Attention Block.* Window attention restricts self-attention to only process a fixed number of neighboring tokens. We can add or remove windows per block to balance accuracy, global information mixing, and computational efficiency.

At inference time, we pick a specific model configuration to select an operating point on the accuracy-latency Pareto curve. Importantly, different model configurations may have similar parameter counts but significantly different latencies. Similar to Cai et al. (2019), we see little benefit from fine-tuning the NAS-mined models on COCO (Appendix **??**), but note modest improvements from fine-tuning NAS-mined models on RF100-VL. This additional fine-tuning is optional, and is often unnecessary for practical deployment. We posit that RF-DETR benefits from additional fine-tuning on RF100-VL because the "architecture augmentation" regularization requires more than 100 epochs to converge on small datasets. Notably, prior weight-sharing NAS methods (Cai et al., 2019) train in stages and use a different learning-rate scheduler per-stage. However, such schedulers make strict assumptions about model convergence, which may not hold across diverse datasets.

**Training Schedulers and Augmentations Bias Model Performance.** State-of-the-art detectors often require careful hyper-parameter tuning to maximize performance on standard benchmarks. However, such bespoke training procedures implicitly bias the model towards certain dataset characteristics (e.g. number of images). Concurrent with DINOv3 (Siméoni et al., 2025), we observe that cosine schedules assume a known (fixed) optimization horizon, which is impractical for diverse target datasets like those in RF100-VL. Data augmentations introduce similar biases by presuming prior knowledge of dataset properties. For example, prior work leverages aggressive data augmentation (e.g., `VerticalFlip`, `RandomFlip`, `RandomResize`, `RandomCrop`, `YOLOXHSVRandomAug`, and `CachedMixUp`) to increase effective dataset size. However, certain augmentations like `VerticalFlip` may negatively bias model predictions in safety-critical domains. For example, a `person` detector in a self-driving vehicle should not be trained with `VerticalFlip` to avoid false positive detections from reflections in puddles. Therefore, we limit augmentations to horizontal flips and random crops. Lastly, LW-DETR applies a per-image random resize augmentation, where each image is padded to match the largest image in the batch. As a result, most images have significant padding, which introduces window artifacts, and wastes computation on padded regions. In contrast, we resize images at the batch level to minimize the number of padded pixels per-batch and to ensure that all positional encoding resolutions are equally likely to be seen at train time.

## 4 EXPERIMENTS

We evaluate RF-DETR on COCO and RF100-VL and demonstrate that our approach achieves state-of-the-art accuracy among all real-time methods. In addition, we identify inconsistencies in standard benchmarking protocols and present a simple standardized procedure to improve reproducibility. Following LW-DETR (Chen et al., 2024a), we group models of similar latency into the same size bucket rather than grouping based on parameter count.

**Datasets and Metrics.** We evaluate RF-DETR on COCO for fair comparison with prior work and on RF100-VL to evaluate generalization to real-world datasets with significantly different data distributions. Due to the diversity of RF100-VL's 100 datasets, we posit that overall performance on this benchmark is a proxy for transferability to any target domain. We use pycocotools to report standard metrics like mean average precision (mAP) and provide breakdown analysis for $AP_{50}$, $AP_{75}$, $AP_{Small}$, $AP_{Medium}$, and $AP_{Large}$. Further, we evaluate efficiency by measuring GFLOPs, number of parameters, and inference latency on an NVIDIA T4 GPU with Tensor-RT 10.4 and CUDA 12.4.

**Standardizing Latency Benchmarking.** Despite its maturity, benchmarking object detectors remains inconsistent across prior work. For example, YOLO-based models often omit non-maximal suppression (NMS) when computing latency, leading to unfair comparisons with end-to-end detec-

Table 1: **Standardizing Latency Evaluation.** Variance in latency measurements can be largely attributed to power throttling and GPU overheating. We mitigate this issue by buffering for 200ms between forward passes. Notably, this benchmarking approach is not designed to measure sustained throughput, but rather ensures reproducible latency measurements. We are unable to reproduce YOLOv8 and YOLOv11's mAP results in TensorRT, likely because these models evaluate with multi-class NMS but only use single-class NMS in inference. We use the standard NMS-tuned confidence threshold of 0.01. YOLOv8 and YOLOv11 performance degrades further when quantizied from FP32 to FP16, reaffirming that all models should report latency and accuracy using the same model artifact. Notably, naively quantizing D-FINE to FP16 reduces performance to 0.5 AP. We fix this issue by changing the authors' export code to use ONNX opset 17 (Appendix **??**).

| Method | Reported | | Buffering (FP-32) | | Buffering (FP-16) | |
|---|---|---|---|---|---|---|
| | $AP_{50:95}$ | Latency (ms) | $AP_{50:95}$ | Latency (ms) | $AP_{50:95}$ | Latency (ms) |
| YOLOv8 (M) | 50.2 | 5.86 | 49.3 | 14.8 | 47.3 | 5.4 |
| YOLOv11 (M) | 51.5 | 4.7 | 49.7 | 18.7 | 48.3 | 5.2 |
| RT-DETR (R18) | 49.0 | 4.61 | 49.0 | 12.2 | 49.0 | 4.4 |
| LW-DETR (M) | 52.5 | 5.6 | 52.6 | 26.8 | 52.6 | 4.4 |
| D-FINE (M) | 55.1 | 5.62 | 55.1 | 13.9 | 55.0 (0.5*) | 5.4 |
| RF-DETR (M) | - | - | 54.8 | 20.5 | 54.7 | 4.4 |

Table 2: **COCO Detection Evaluation.** We compare RF-DETR with popular real-time and open-vocabulary object detectors below. We find that RF-DETR (nano) outperforms D-FINE (nano) and LW-DETR (tiny) by more than 5 AP. RF-DETR significantly outperforms YOLOv8 and YOLOv11, while RF-DETR's nano size achieves performance parity with YOLOv8 and YOLOv11's medium size model. We denote models that do not support TensorRT execution with a star, and instead report PyTorch latency results. See Appendix **??** for L, XL, and Max variants of RF-DETR on COCO.

| Model | Size | # Params. | GFLOPS | Latency (ms) | AP | $AP_{50}$ | $AP_{75}$ | $AP_S$ | $AP_M$ | $AP_L$ |
|---|---|---|---|---|---|---|---|---|---|---|
| **Real-Time Object Detection w/ NMS** | | | | | | | | | | |
| YOLOv8 (Jocher et al., 2023) | N | 3.2M | 8.7 | 2.1 | 35.2 | 49.2 | 38.3 | 15.8 | 38.8 | 51.3 |
| YOLOv11 (Jocher et al., 2024) | N | 2.6M | 6.5 | 2.2 | 37.1 | 51.6 | 40.4 | 17.3 | 40.7 | 55.6 |
| YOLOv8 (Jocher et al., 2023) | S | 11.2M | 28.6 | 2.9 | 42.4 | 57.6 | 46.0 | 22.2 | 47.1 | 59.6 |
| YOLOv11 (Jocher et al., 2024) | S | 9.4M | 21.5 | 3.2 | 44.1 | 59.3 | 47.9 | 26.1 | 48.5 | 62.6 |
| YOLOv8 (Jocher et al., 2023) | M | 25.9M | 78.9 | 5.4 | 47.3 | 62.5 | 51.5 | 27.5 | 52.9 | 65.1 |
| YOLOv11 (Jocher et al., 2024) | M | 20.1M | 68.0 | 5.1 | 48.3 | 63.6 | 52.5 | 29.1 | 53.8 | 66.3 |
| **Open-Vocabulary Object Detection (Fully-Supervised Fine-Tuning)** | | | | | | | | | | |
| GroundingDINO (Liu et al., 2023) | T | 173.0M | 1008.3 | 427.6* | 58.2 | - | - | - | - | - |
| **End-to-End Real-Time Object Detection** | | | | | | | | | | |
| LW-DETR (Chen et al., 2024a) | T | 12.1M | 21.4 | 1.9 | 42.9 | 60.7 | 45.9 | 22.7 | 47.3 | 60.0 |
| D-FINE (Peng et al., 2024) | N | 3.8M | 7.3 | 2.1 | 42.7 | 60.2 | 45.4 | 22.9 | 46.6 | 62.1 |
| RF-DETR (Ours) | N | 30.5M | 31.9 | 2.3 | 48.0 | 67.0 | 51.4 | 25.2 | 53.5 | 70.0 |
| LW-DETR (Chen et al., 2024a) | S | 14.6M | 31.8 | 2.6 | 48.0 | 66.8 | 51.6 | 26.7 | 52.5 | 65.6 |
| D-FINE (Peng et al., 2024) | S | 10.2M | 25.2 | 3.5 | 50.6 | 67.6 | 55.0 | 32.6 | 54.6 | 66.6 |
| RF-DETR (Ours) | S | 32.1M | 59.8 | 3.5 | 52.9 | 71.9 | 57.0 | 32.0 | 58.3 | 73.0 |
| RT-DETR (Zhao et al., 2024) | R18 | 36.0M | 100.0 | 4.4 | 49.0 | 66.6 | 53.3 | 32.8 | 52.1 | 65.0 |
| LW-DETR (Chen et al., 2024a) | M | 28.2M | 83.9 | 4.4 | 52.6 | 72.0 | 56.6 | 32.5 | 57.6 | 70.5 |
| D-FINE (Peng et al., 2024) | M | 19.2M | 56.6 | 5.4 | 55.0 | 72.6 | 59.7 | 37.6 | 59.4 | 71.7 |
| RF-DETR (Ours) | M | 33.7M | 78.8 | 4.4 | 54.7 | 73.5 | 59.2 | 36.1 | 59.7 | 73.8 |
| RF-DETR (Ours) | 2XL | 126.9M | 438.4 | 17.2 | 60.1 | 78.5 | 65.5 | 43.2 | 64.9 | 76.2 |

tors. Additionally, YOLO-based segmentation models measure the latency of generating prototype predictions instead of directly usable per-object masks (Jocher et al., 2024), leading to biased run-time measurements. Further, D-FINE's reported latency evaluation of LW-DETR is 25% faster than reported by Chen et al. (2024b). We observe that such differences can be attributed to detectable power throttling events, particularly when the GPU overheats (Table 1). In contrast, simply pausing for 200ms between consecutive forward passes largely mitigates power throttling, yielding more stable latency measurements (Appendix **??**). Lastly, we find that prior work often reports latency using FP16 quantized models, but evaluates accuracy with FP32 models. However, naive quantization can significantly degrade performance (in some cases dropping performance to near 0 AP). To ensure fair comparison, we advocate for reporting accuracy and latency with the same model artifact. We release our stand-alone benchmarking tool on GitHub.

**Evaluating RF-DETR and RF-DETR-Seg on COCO.** COCO (Lin et al., 2014) is a flagship benchmark for object detection and instance segmentation. In Table 2, we compare RF-DETR with leading real-time and open-vocabulary detectors. RF-DETR (nano) beats both D-FINE (nano) and LW-DETR (nano) by more than 5 AP. We see similar trends for small and medium sizes as well. No-

Table 3: **COCO Instance Segmentation Evaluation.** We compare RF-DETR with popular real-time instance segmentation methods on COCO. Notably, RF-DETR (nano) outperforms all reported YOLOv8 and YOLOv11 model sizes. Further RF-DETR (nano) outperforms FastInst by 5.4%, while running nearly ten times faster. RF-DETR (medium) approaches the performance on MaskDINO at a fraction of the runtime. We denote models that do not support TensorRT execution with a star, and instead report PyTorch latency results. Our latencies for YOLOs also include the conversion of protos into masks, which are not typically included in prior benchmarks but nonetheless contribute meaningfully to practical latency. See Appendix **??** for L, XL, and Max variants of RF-DETR-Seg on COCO.

| Model | Size | # Params. | GFLOPS | Latency (ms) | AP | $AP_{50}$ | $AP_{75}$ | $AP_S$ | $AP_M$ | $AP_L$ |
|---|---|---|---|---|---|---|---|---|---|---|
| **Real-Time Instance Segmentation w/ NMS** | | | | | | | | | | |
| YOLOv8 (Jocher et al., 2023) | N | 3.4M | 12.6 | 3.5 | 28.3 | 45.6 | 29.8 | 9.3 | 31.3 | 44.3 |
| YOLOv11 (Jocher et al., 2024) | N | 2.9M | 10.4 | 3.6 | 30.0 | 47.8 | 31.5 | 10.0 | 33.4 | 47.7 |
| YOLOv8 (Jocher et al., 2023) | S | 11.8M | 42.6 | 4.2 | 34.0 | 53.8 | 36.0 | 13.6 | 38.5 | 52.2 |
| YOLOv11 (Jocher et al., 2024) | S | 10.1M | 35.5 | 4.6 | 35.0 | 55.4 | 37.1 | 15.3 | 39.7 | 53.9 |
| YOLOv8 (Jocher et al., 2023) | M | 27.3M | 110.2 | 7.0 | 37.3 | 58.2 | 39.9 | 16.7 | 43.0 | 56.1 |
| YOLOv11 (Jocher et al., 2024) | M | 22.4M | 123.3 | 6.9 | 38.5 | 60.0 | 40.9 | 18.0 | 44.3 | 57.6 |
| **End-to-End Instance Segmentation** | | | | | | | | | | |
| RF-DETR-Seg. (Ours) | N | 33.6M | 50.0 | 3.4 | 40.3 | 63.0 | 42.6 | 16.3 | 45.3 | 63.6 |
| RF-DETR-Seg. (Ours) | S | 33.7M | 70.6 | 4.4 | 43.1 | 66.2 | 45.9 | 21.9 | 48.5 | 64.1 |
| FastInst (He et al., 2023) | R50 | 29.7M | 99.7 | 39.6* | 34.9 | 56.0 | 36.2 | 13.3 | 38.0 | 56.8 |
| MaskDINO (Li et al., 2023) | R50 | 52.1M | 586 | 242* | 46.3 | 69.0 | 50.7 | 26.1 | 49.3 | 66.1 |
| RF-DETR-Seg. (Ours) | M | 35.7M | 102.0 | 5.9 | 45.3 | 68.4 | 48.8 | 25.5 | 50.4 | 65.3 |
| RF-DETR (Ours) | 2XL | 38.6M | 435.3 | 21.8 | 49.9 | 73.1 | 54.5 | 33.9 | 54.1 | 65.7 |

tably, RF-DETR also significantly outperforms YOLOv8 and YOLOv11. RF-DETR (nano) matches the performance of YOLOv8 and YOLOv11 (medium). We use mmdetection's implementation of GroundingDINO and include their reported AP since they do not release a model artifact for GroundingDINO fine-tuned on COCO. We benchmark mmGroundingDINO's parameter count, GFLOPS, and latency using the released open-vocabulary model. In Table 3, we compare RF-DETR-Seg with real-time instance segmentation models. RF-DETR-Seg (nano) outperforms YOLOv8 and YOLOv11 at all sizes. Furthermore, RF-DETR-Seg (nano) beats FastInst by 5.4% while running almost ten times faster. Similarly, RF-DETR (x-large) surpasses GroundingDINO (tiny), and RF-DETR-Seg (large) outperforms MaskDINO (R50), at a fraction of their runtime.

**Evaluating RF-DETR on RF100-VL.** RF100-VL is a challenging detection benchmark composed of 100 diverse datasets. We report latencies, FLOPs, and accuracy averaged over all 100 datasets in Table 4. Our results show that RF-DETR (2x-large) outperforms GroundingDINO and LLMDet while requiring only a fraction of their runtime. Interestingly, RT-DETR outperforms D-FINE (which is built on RT-DETR) at $AP_{50}$, indicating that D-FINE's hyperparameters are potentially overfit to COCO. We also note that RF-DETR benefits from scaling to larger backbone sizes (Appendix **??**). In contrast, YOLOv8 and YOLOv11 consistently underperform DETR-based detectors, and scaling these model families to larger sizes does not improve their performance on RF100-VL (Figure 1).

**Impact of Neural Architecture Search.** We ablate the impact of weight-sharing NAS in Table 3. We find that adopting a gentler set of hyperparameters compared to LW-DETR (e.g. larger batch size, lower learning rate, and replacing batch normalization with layer normalization) reduces performance over LW-DETR by 1.0%. Notably, replacing batch normalization with layer normalization hurts performance, but is necessary to train on consumer hardware. However, replacing LW-DETR's CAEv2 backbone with DINOv2 improves performance by 2%. The lower learning rate, in particular, helps preserve DINOv2's pre-trained knowledge, while additional epochs of Objects-365 pre-training further compensate for the slower optimization. Our final model with weight-sharing NAS improves over LW-DETR by 2% without increasing latency.

**Impact of Backbone Architecture and Pre-Training.** We study the impact of different backbone architectures in RF-DETR. We find that DINOv2 achieves the best performance, outperforming CAEv2 by 2%. Interestingly, despite having fewer parameters than SigLIPv2, SAM2's Hiera-S backbone is considerably slower. This is in contrast with Hiera's claim that it is meaningfully faster than equivalently performant ViTs. However, Hiera does not explore latency in the context of Flash Attention kernels, which are highly optimized in compilers such as TensorRT. Additionally, existing

Table 4: **RF100-VL Evaluation.** We compare RF-DETR with real-time and open-vocabulary object detectors on RF100-VL. Interestingly, RF-DETR (2x-large) outperforms GroundingDINO (tiny), and LLMDet (tiny) at a fraction of their runtime. We report the average latency and FLOPs over all 100 datasets. We note that YOLOv8 and YOLOv11's latency measurements may be suboptimal because the default tuned NMS threshold of 0.01 may not work well for all datasets in RF100-VL. We denote models that do not support TensorRT execution with a star, and instead report PyTorch latency results. See Appendix **??** for L, XL, and Max variants of RF-DETR on RF100-VL.

| Model | Size | # Params. | GFLOPS | Latency (ms) | AP | $AP_{50}$ | $AP_{75}$ | $AP_S$ | $AP_M$ | $AP_L$ |
|---|---|---|---|---|---|---|---|---|---|---|
| **Real-Time Object Detectors w/ NMS** | | | | | | | | | | |
| YOLOv8 (Jocher et al., 2023) | N | 3.2M | 8.7 | 2.6 | 55.0 | 81.1 | 59.5 | 4.8 | 44.1 | 48.0 |
| YOLOv11 (Jocher et al., 2024) | N | 2.6M | 6.5 | 3.0 | 55.5 | 81.3 | 60.3 | 4.7 | 44.4 | 49.2 |
| YOLOv8 (Jocher et al., 2023) | S | 11.2M | 28.6 | 3.1 | 56.3 | 82.0 | 60.9 | 6.1 | 45.6 | 48.6 |
| YOLOv11 (Jocher et al., 2024) | S | 9.4M | 21.5 | 3.3 | 56.4 | 82.5 | 61.3 | 6.5 | 45.5 | 48.5 |
| YOLOv8 (Jocher et al., 2023) | M | 25.9M | 78.9 | 5.4 | 56.5 | 82.3 | 60.9 | 6.4 | 45.7 | 48.6 |
| YOLOv11 (Jocher et al., 2024) | M | 20.1M | 68.0 | 5.1 | 57.0 | 82.5 | 61.9 | 7.3 | 46.1 | 48.6 |
| **Open-Vocabulary Object-Detectors (Fully-Supervised Fine-Tuning)** | | | | | | | | | | |
| GroundingDINO (Liu et al., 2023) | T | 173.0M | 1008.3 | 309.9* | 62.3 | 88.8 | 67.8 | 39.2 | 57.7 | 69.5 |
| LLMDet (Fu et al., 2025) | T | 173.0M | 1008.3 | 308.4* | 62.3 | 88.3 | 67.8 | 39.1 | 57.6 | 70.3 |
| **End-to-End Real-Time Object Detectors** | | | | | | | | | | |
| LW-DETR (Chen et al., 2024a) | N | 12.1M | 21.4 | 1.9 | 57.1 | 84.7 | 61.5 | 31.2 | 51.8 | 65.8 |
| D-FINE (Peng et al., 2024) | N | 3.8M | 7.3 | 2.0 | 58.2 | 84.4 | 62.5 | 32.4 | 52.9 | 65.8 |
| RF-DETR (Ours) | N | 31.2M | 34.5 | 2.5 | 57.8 | 85.5 | 61.5 | 30.1 | 52.2 | 67.2 |
| RF-DETR w/ Fine-Tuning (Ours) | N | 31.2M | 34.5 | 2.5 | 58.6 | 85.7 | 63.0 | 31.0 | 53.2 | 67.6 |
| LW-DETR (Chen et al., 2024a) | S | 14.6M | 31.8 | 2.6 | 57.4 | 85.0 | 62.0 | 32.1 | 52.1 | 65.8 |
| D-FINE (Peng et al., 2024) | S | 10.2M | 25.2 | 3.5 | 60.3 | 85.3 | 65.4 | 36.6 | 56.0 | 68.4 |
| RF-DETR (Ours) | S | 33.5M | 62.4 | 3.7 | 60.9 | 87.5 | 66.1 | 34.2 | 55.7 | 69.6 |
| RF-DETR w/ Fine-Tuning (Ours) | S | 33.5M | 62.4 | 3.7 | 61.2 | 87.7 | 66.1 | 34.9 | 55.6 | 69.5 |
| RT-DETR (Zhao et al., 2024) | M | 36.0M | 100.0 | 4.3 | 59.6 | 85.7 | 64.6 | 36.4 | 54.6 | 67.3 |
| LW-DETR (Chen et al., 2024a) | M | 28.2M | 83.9 | 4.3 | 59.8 | 86.8 | 64.9 | 34.0 | 54.4 | 68.9 |
| D-FINE (Peng et al., 2024) | M | 19.2M | 56.6 | 5.6 | 60.6 | 85.5 | 65.8 | 36.0 | 56.6 | 67.5 |
| RF-DETR (Ours) | M | 33.5M | 86.7 | 4.6 | 61.7 | 88.0 | 66.9 | 35.8 | 56.5 | 70.0 |
| RF-DETR w/ Fine-Tuning (Ours) | M | 33.5M | 86.7 | 4.6 | 62.0 | 88.1 | 67.1 | 36.2 | 56.4 | 70.2 |
| RF-DETR (Ours) | 2XL | 123.5M | 410.2 | 15.6 | 63.3 | 88.9 | 69.0 | 38.7 | 58.2 | 71.6 |
| RF-DETR (Ours) w/ Fine-Tuning | 2XL | 123.5M | 410.2 | 15.6 | 63.5 | 89.0 | 69.2 | 38.9 | 58.3 | 71.7 |

Table 5: **Ablation on Neural Architecture Search.** We ablate the impact of each "tunable knob" on accuracy and latency below. Using a gentler set of hyperparameters compared to LW-DETR (e.g. smaller batch size, lower learning rate, replacing batch norm with layer norm) reduces performance by 1%. However, we regain this lost performance by replacing LW-DETR's CAEV2 backbone with DINOv2. Importantly, the lower learning rate and layer-norm allow us to better preserve DINOv2's foundational knowledge and allows us to train with larger batch sizes, making weight-sharing NAS more effective. Counterintuitively, introducing weight sharing NAS to the training scheme improves the performance of the base configuration even though patch size 14 isn't in the NAS search space.

| Model | # Params. | GFLOPS | Latency (ms) | AP | $AP_{50}$ | $AP_{75}$ | $AP_S$ | $AP_M$ | $AP_L$ |
|---|---|---|---|---|---|---|---|---|---|
| LW-DETR (M) | 28.2M | 83.7 | 4.4 | 52.6 | 72.0 | 56.6 | 32.5 | 57.6 | 70.5 |
| + Gentler Hyperparameters | 28.2M | 83.7 | 4.4 | 51.6 | 71.1 | 55.5 | 31.7 | 56.4 | 69.4 |
| + DINOv2 Backbone | 32.3M | 78.2 | 4.7 | 53.6 | 72.7 | 58.0 | 34.3 | 58.3 | 72.4 |
| + Additional O365 Pre-Training | 32.3M | 78.2 | 4.7 | 54.3 | 73.4 | 58.8 | 35.8 | 59.2 | 72.3 |
| + Weight Sharing NAS | 32.3M | 78.2 | 4.7 | 54.6 | 73.4 | 59.3 | 36.3 | 59.3 | 72.1 |
| + Patch Size $14 \to 16$, Res $560 \to 640$ | 32.3M | 78.5 | 4.7 | 54.4 | 73.2 | 59.1 | 35.9 | 59.2 | 72.1 |
| + Image Resolution $640 \to 576$ | 32.2M | 64.2 | 4.0 | 53.6 | 72.4 | 58.2 | 34.8 | 58.6 | 72.0 |
| + # Windows per Block $4 \to 2$ | 32.2M | 63.7 | 4.3 | 54.3 | 73.3 | 58.8 | 35.6 | 59.4 | 73.2 |
| + # Decoder Layers $3 \to 4$ | 33.7M | 64.8 | 4.4 | 54.6 | 73.5 | 59.1 | 36.0 | 59.8 | 73.7 |
| + # Query Tokens $300 \to 300$ | 33.7M | 64.8 | 4.4 | 54.6 | 73.5 | 59.1 | 36.0 | 59.8 | 73.7 |

foundation model families typically do not release lightweight ViT variants such as ViT-S or ViT-T, making it difficult to repurpose such models for real-time applications.

**Rethinking Standard Accuracy Benchmarking Practices.** Following prior work, we report all COCO results on the validation set. However, relying solely on the validation for both model selection and evaluation can lead to overfitting. For example, D-FINE (which builds on RT-DETR) conducts an extensive hyperparameter sweep on COCO's validation set and reports its best model. However, evaluating this configuration on RF100-VL shows that D-FINE underperforms RT-DETR on the test set. In contrast, our method achieves state-of-the-art performance among all real-time detectors on both RF100-VL and COCO, demonstrating the robustness of our weight-sharing NAS. In

Table 6: **Ablation on Backbone.** We ablate the impact of using different backbone architectures for RF-DETR below. We find that DINOv2 achieves the highest performance, outperforming CAEv2 by 2.4%. All models are pretrained with 60 epochs of Objects-365 and the "Gentler Hyperparameters" setting. Note that SAM2 and SigLIPv2 perform poorly when evaluated in FP16. Therefore, we report FP16 TensorRT latency with FP32 ONNX accuracy for these two models as an upper bound on their performance if optimized for FP16.

| LW-DETR (M) + Gentler Hyperparameters | # Params. | GFLOPS | Latency (ms) | AP | $AP_{50}$ | $AP_{75}$ | $AP_S$ | $AP_M$ | $AP_L$ |
|---|---|---|---|---|---|---|---|---|---|
| w/ CAEv2 ViT/S-16-Truncated Backbone | 28.3M | 83.7 | 4.4 | 52.3 | 71.4 | 56.3 | 32.3 | 56.4 | 70.0 |
| w/ DINOv2 ViT/S-14 Backbone | 32.3M | 78.2 | 4.7 | 54.3 | 73.4 | 58.8 | 35.8 | 59.2 | 72.3 |
| w/ SigLIPv2 ViT/B-32 Backbone* | 105.1M | 81.6 | 4.8 | 50.4 | 70.4 | 53.7 | 28.0 | 55.3 | 73.0 |
| w/ SAM2 Hiera-S Backbone* | 44.0M | 109.1 | 11.2 | 53.6 | 72.4 | 57.9 | 33.3 | 58.3 | 71.0 |

addition to evaluating on COCO, we advocate that future detectors should also evaluate on datasets with public validation and test splits like RF100-VL.

**Limitations.** Despite controlling for power throttling and GPU overheating during inference, our latency measurements still have a variance of up to 0.1ms due to the non-deterministic behavior of TensorRT during compilation. Specifically, TensorRT can introduce power throttling, which in turn affects the resulting engine and leads to random fluctuations in latency. Although the measurement of a given TensorRT engine is generally consistent, recompiling the same ONNX artifact can produce slightly different latency results. Therefore, we only report latencies with one digit of precision after the decimal place.

## 5 CONCLUSION

In this paper, we introduce RF-DETR, a state-of-the-art NAS-based method for fine-tuning specialist end-to-end object detectors for diverse target datasets and hardware platforms. Our approach outperforms prior state-of-the-art real-time methods on COCO and RF100-VL, improving upon D-FINE (nano) by 5% AP on COCO. Moreover, we highlight that current architectures, learning rate schedulers and augmentation schedulers are tailored to maximize performance on COCO, suggesting that the community should benchmark models on diverse, large-scale datasets to prevent implicit overfitting. Lastly, we highlight the high variance in latency benchmarking due to power throttling and propose a standardized protocol to improve reproducibility.

## ACKNOWLEDGEMENTS

This work was supported in part by compute provided by NVIDIA DGX. We'd like to thank Brad Dwyer for reviewing early versions of our manuscript.

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
