## A    IMPLEMENTATION DETAILS

**Training Hyperparamters.** RF-DETR extends LW-DETR (Chen et al., 2024a) for Neural Architecture Search. We highlight key differences in our training procedure below. First, we pseudo-label Objects-365 (Shao et al., 2019) with SAM2 (Ravi et al., 2024) to allow us to pre-train the segmentation and detection heads on the same data. We use a learning rate of 1e-4 (LW-DETR uses 4e-4), and a batch size of 128 (LW-DETR uses the same). Similar to DINOv3 (Siméoni et al., 2025), we use an EMA scheduler since this is necessary for EMA to function. However, unlike DINOv3, we omit learning-rate warm-up. We clip all gradients greater than 0.1 and apply a per-layer multiplicative decay of 0.8 to preserve information (especially the earlier layers) in the DINOv2 backbone. We place our window attention blocks between layers {0, 1, 3, 4, 6, 7, 9, 10}, while LW-DETR places their window attention blocks between layers {0, 1, 3, 6, 7, 9}. Although we have the same number of windows, contiguous windowed blocks don't require an additional reshape operation, making our implementation slightly more efficient. Further, we train with more multi-scale resolutions (0.5 to 1.5 scale) than LW-DETR (0.7 to 1.4 scale) to ensure that the augmentation is symmetric around the default scale. Notably, we add resolution as a "tunable knob" in our NAS search space, while LW-DETR uses it as a form of data augmentation. Our model training and inference code is available on GitHub.

**Latency Evaluation.** We ensure fair evaluation between models by measuring detection accuracy and latency using the same artifact. To further standardize inference, we employ CUDA graphs in TensorRT, which pre-queue all kernels rather than requiring the CPU to launch them serially during execution. This optimization can accelerate some networks depending on the number and type of kernels used by the model. We observe that RT-DETR, LW-DETR, and RF-DETR benefit from this optimization. Further, CUDA graphs place LW-DETR on the same latency-accuracy curve as D-FINE, since CUDA graphs speed up LW-DETR but do not benefit D-FINE. We release our stand-alone latency benchmarking tool on GitHub.

**Pareto-Optimal Model Configurations in COCO.** We present the Pareto-Optimal RF-DETR and RF-DETR-Seg configurations in Tables 7 and 8. We highlight notable trends about RF-DETR's Pareto-Optimal architectures in Appendix L.

Table 7: **RF-DETR COCO Detection Model Config.**

| Model Size | Resolution | Patch Size | Windows | Decoder Layers | Queries | Backbone |
|---|---|---|---|---|---|---|
| N | 384 | 16 | 2 | 2 | 300 | DINOv2-S |
| S | 512 | 16 | 2 | 3 | 300 | DINOv2-S |
| M | 576 | 16 | 2 | 4 | 300 | DINOv2-S |
| L | 704 | 16 | 2 | 4 | 300 | DINOv2-S |
| XL | 700 | 20 | 1 | 5 | 300 | DINOv2-B |
| 2XL | 880 | 20 | 2 | 5 | 300 | DINOv2-B |
| Max | 828 | 12 | 1 | 6 | 300 | DINOv2-B |

Table 8: **RF-DETR-Seg COCO Segmentation Model Config.**

| Model Size | Resolution | Patch Size | Windows | Decoder Layers | Queries | Backbone |
|---|---|---|---|---|---|---|
| N | 312 | 12 | 1 | 4 | 100 | DINOv2-S |
| S | 384 | 12 | 2 | 4 | 100 | DINOv2-S |
| M | 432 | 12 | 2 | 5 | 200 | DINOv2-S |
| L | 504 | 12 | 2 | 5 | 300 | DINOv2-S |
| XL | 624 | 12 | 2 | 6 | 300 | DINOv2-S |
| 2XL | 768 | 12 | 2 | 6 | 300 | DINOv2-S |
| Max | 890 | 10 | 1 | 6 | 300 | DINOv2-S |

**Parameter Sampling Grid.** Lastly, we present our sampling grid for training and inference below. Importantly, we only drop decoder layers and queries during inference. We uniformly sample configurations during training and perform grid search over all configurations during inference to find Pareto-Optimal model configurations. RF-DETRs total training time is roughly two to four times as long as a non-NAS baseline, depending on the target dataset. However, RF-DETR can generate all size configurations from this single training run, while other non-NAS baselines must be re-trained for each new model size. We evaluate 6,468 network configurations (11 resolutions * 7 patch sizes * 7 decoder layers * 3 windows * 4 query settings) during architecture search. We estimate that this search takes approximately 10,000 GPU Hours (200 T4 GPUs * 48 hours).

**Training Configurations**

- Image Resolutions: 320, 384, 448, 512, 576, 640, 704, 768, 832, 896, 960
- Patch Sizes: 8, 10, 12, 16, 20, 24, 32
- Number of Decoder Layers: 6
- Number of Windows: 1, 2, 4
- Number of Queries: 300

**Inference Configurations**

- Image Resolutions: 320, 384, 448, 512, 576, 640, 704, 768, 832, 896, 960
- Patch Sizes: 8, 10, 12 ,14, 16, 20, 24, 32
- Number of Decoder Layers: 0, 1, 2, 3, 4, 5, 6
- Number of Windows: 1, 2, 4
- Number of Queries: 50, 100, 200, 300

## B   ABLATION ON QUERY TOKENS AND DECODER LAYERS

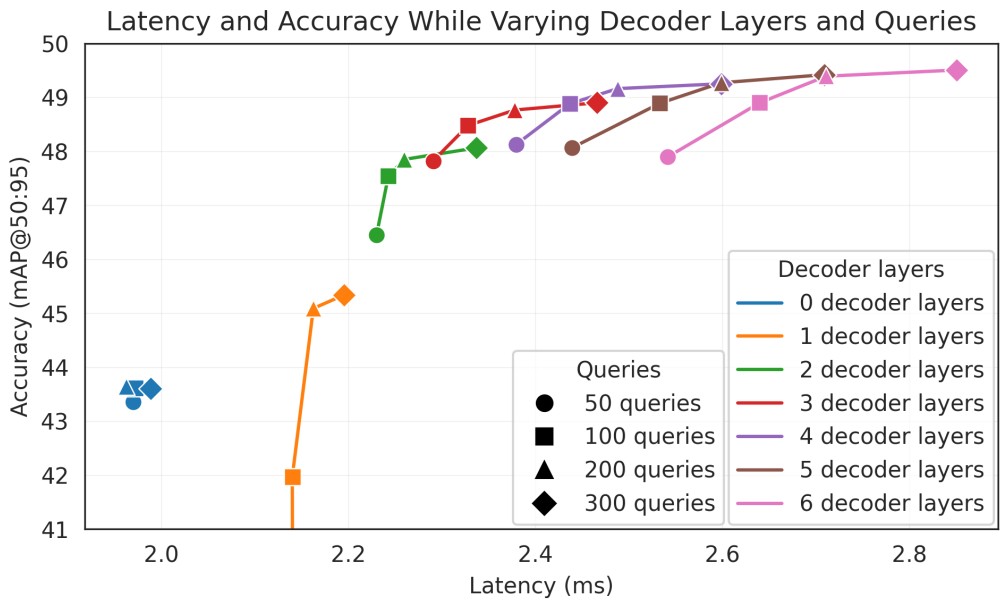

Figure 4: **Impact of Decoder Layers vs. Query Tokens**. We evaluate the impact of inference-time query dropping for trading-off accuracy and latency in RF-DETR (nano). Interestingly, we find that dropping the 100 lowest confidence queries does not significantly reduce performance, but modestly improves latency for all decoder layers.

We train RF-DETR (nano) with 300 object queries, following standard practices for real-time DETR-based object detectors. However, many datasets contain fewer than 300 objects per image. Therefore, processing all 300 queries can be computationally wasteful. LW-DETR (tiny) demonstrates that training with fewer queries can improve the latency-accuracy tradeoff. Rather than deciding on the optimal number of queries apriori, we find that we can drop queries at test time *without retraining* by discarding the lowest-confidence queries ordered by the confidence of the corresponding token at the output of the encoder. As shown in Figure 4, this yields meaningful latency-accuracy tradeoffs. In addition, prior work (Zhao et al., 2024) demonstrates that decoder layers can be pruned at test time, since each layer is supervised independently during training. We

find that it is possible to remove *all* decoder layers, relying solely on the initial query proposals from the two-stage DETR pipeline. In this case, there is no cross-attention to the encoder states or self-attention between queries, leading to a substantial runtime reduction. The resulting model resembles a single-stage YOLO-style architecture without NMS. Eliminating the final decoder layer reduces latency by 10% with only a 2 mAP drop in performance.

## C    BENCHMARKING FLOPs

We benchmark FLOPs for RF-DETR, GroundingDINO, and LLMDet with PyTorch's `FlopCounterMode`. We find that `FlopCounterMode` closely reproduces FLOP counts obtained with custom benchmarking tools for YOLOv11, D-FINE, and LW-DETR. In practice, we also find that it provides more reliable results than CalFLOPs (Ye, 2023). Notably, LW-DETR's FLOPs count is roughly twice that of the originally reported result (cf. Table 9). We posit that this discrepancy can be attributed to LW-DETR reporting MACs instead of FLOPs. We rely on the officially reported FLOPs counts from YOLOv11, YOLOv8, D-FINE, and RT-DETR.

Table 9: **FLOPs Benchmarking Comparison.** We compare FLOPs reported with custom benchmarking tools, CalFLOPs, and PyTorch's FlopCounterMode. Notably, we find that FlopCounter-Model closely matches the results reported with custom benchmarking code, suggesting that it is more reliable than prior generic benchmarking tools.

| Model | Size | Reported | CalFLOPs | FlopCounterMode |
|---|---|---|---|---|
| D-FINE | S | 25.2 M | 25.2 M | 25.5 M |
| LW-DETR | S | 16.6 M | 22.9 M | 31.8 M |
| YOLO11 | S | 21.5 M | 23.9 M | 21.6 M |

## D    IMPACT OF CLASS-NAMES ON OPEN-VOCABULARY DETECTORS

We evaluate the impact of fine-tuning open-vocabulary detectors like GroundingDINO with class names on RF100-VL in Table 10. Intuitively, GroundingDINO's vision-language pre-training should be more more useful when we prompt with class names (e.g. `car`, `truck`, `bus`) instead of class indices (e.g. 0, 1, 2). Using class names when fine-tuning provides more information to the VLM about the underlying data than is available to non-VLM detectors, potentially leading to better downstream performance. However, we find that fine-tuning GroundingDINO on RF100-VL yields nearly identical performance in both cases, suggesting that naively fine-tuning the end-to-end model mitigates the benefits of open-vocabulary pre-training. Future work should investigate ways of effectively fine-tuning VLMs to preserve foundational pre-training.

Table 10: **Evaluating the Impact of Class Names.** We evaluate the impact of using class-names when fine-tuning VLMs like GroundingDINO. We find that class-names do not provide significant benefit over prompting with class indices, suggesting that fine-tuning has diminished the impact of internet-scale pre-training.

| Model | Size | # Params. | GFLOPS | Latency (ms) | AP | $AP_{50}$ | $AP_{75}$ | $AP_S$ | $AP_M$ | $AP_L$ |
|---|---|---|---|---|---|---|---|---|---|---|
| **RF100-VL** | | | | | | | | | | |
| GroundingDINO (Liu et al., 2023) w/ Standard Class Names | T | 173.0M | 1008.3 | 309.9* | 62.3 | 88.8 | 67.8 | 39.2 | 57.7 | 69.5 |
| GroundingDINO (Liu et al., 2023) w/ Class Index Names | T | 173.0M | 1008.3 | 309.9* | 62.5 | 88.2 | 68.3 | 40.0 | 58.4 | 70.3 |

## E    BENCHMARKING LARGER MODEL VARIANTS

Detectors like LW-DETR (Chen et al., 2024a) and D-FINE (Peng et al., 2024) hand-design larger variants to scale up a model family. In contrast, NAS-based architectures like RF-DETR automatically discover scaling strategies through grid-based search. We analyze two families of RF-DETR models derived from distinct scaling strategies: one based on a DINOv2-S backbone and another based on a DINOv2-B backbone. To evaluate how well each family scales, we compare their NAS-generated Pareto curves against those of D-FINE. Specifically, at each D-FINE size, we identify the RF-DETR variant with the same backbone that maximizes performance at a comparable latency. For example, when comparing to D-FINE (small), we select the RF-DETR model that offers the

Table 11: **mAP@50:95 Gap of RF-DETR vs D-FINE at Similar Latencies** We compare how different RF-DETR model families scale relative to D-FINE. D-FINE (nano) is excluded since it was not pretrained on Objects-365 and is therefore not expected to follow similar scaling trends. For each RF-DETR backbone, we select the highest accuracy Pareto-optimal NAS-mined model with latency up to that of the corresponding D-FINE variant. Notably, RF-DETR (DINOv2-B) achieves better scalability than RF-DETR (DINOv2-S) and D-FINE. Note that none of the RF-DETR models for COCO are finetuned.

| Method (Backbone) | S | M | L | XL |
|---|---|---|---|---|
| D-FINE (Peng et al., 2024) | 50.6 | 55.4 | 57.2 | 59.3 |
| RF-DETR (DINOv2-S) | +2.3 | +0.9 | -0.4 | -1.1 |
| RF-DETR (DINOv2-B) | -3.1 | -1.3 | -1.2 | -0.7 |

Table 12: **COCO Detection Evaluation for Larger Model Variants.** We present RF-DETR's performance for L, XL, and 2XL sizes on COCO below. Notably, D-FINE (x-large) outperforms RF-DETR (x-large) on mAP 50:95. However, RF-DETR (2x-large) beats D-FINE by 0.8 AP, and is the first real-time detector to surpass 60 AP on COCO.

| Model | Size | # Params. | GFLOPS | Latency (ms) | AP | $AP_{50}$ | $AP_{75}$ | $AP_S$ | $AP_M$ | $AP_L$ |
|---|---|---|---|---|---|---|---|---|---|---|
| **Real-Time Object Detection w/ NMS** | | | | | | | | | | |
| YOLOv8 (Jocher et al., 2023) | L | 43.7M | 165.2 | 8.0 | 49.5 | 64.7 | 54.0 | 30.2 | 55.1 | 68.5 |
| YOLOv11 (Jocher et al., 2024) | L | 25.3M | 86.9 | 6.5 | 49.9 | 64.9 | 54.5 | 30.4 | 55.9 | 68.1 |
| YOLOv8 (Jocher et al., 2023) | XL | 68.2M | 257.8 | 11.3 | 50.5 | 65.6 | 55.1 | 30.0 | 56.2 | 69.5 |
| YOLOv11 (Jocher et al., 2024) | XL | 56.9M | 194.9 | 10.5 | 50.9 | 66.1 | 55.4 | 31.5 | 56.6 | 68.7 |
| **End-to-End Real-Time Object Detection** | | | | | | | | | | |
| RT-DETR (Zhao et al., 2024) | R50 | 42M | 136 | 8.5 | 55.0 | 73.3 | 59.8 | 37.9 | 59.7 | 71.6 |
| LW-DETR (Chen et al., 2024a) | L | 46.8M | 137.5 | 6.9 | 56.1 | 74.6 | 61.0 | 37.1 | 60.4 | 73.0 |
| D-FINE (Peng et al., 2024) | L | 31M | 91 | 7.5 | 57.2 | 74.9 | 62.2 | 40.6 | 61.4 | 73.7 |
| RF-DETR (Ours) | L | 33.9M | 125.6 | 6.8 | 56.5 | 75.1 | 61.3 | 39.0 | 61.0 | 73.9 |
| RT-DETR (Zhao et al., 2024) | R101 | 76M | 259 | 12.0 | 56.1 | 74.5 | 61.1 | 38.1 | 60.4 | 73.4 |
| LW-DETR (Chen et al., 2024a) | XL | 118.0M | 342.5 | 13.0 | 58.3 | 76.9 | 63.3 | 40.2 | 63.3 | 74.7 |
| D-FINE (Peng et al., 2024) | XL | 62M | 202 | 11.5 | 59.3 | 76.8 | 64.6 | 42.1 | 64.2 | 76.3 |
| RF-DETR (Ours) | XL | 126.4M | 299.3 | 11.5 | 58.6 | 77.4 | 63.8 | 40.3 | 63.9 | 76.2 |
| RF-DETR (Ours) | 2XL | 126.9M | 438.4 | 17.2 | 60.1 | 78.5 | 65.5 | 43.2 | 64.9 | 76.2 |
| RF-DETR (Ours) | Max | 132.4M | 1742.5 | 98.0 | 61.8 | 79.7 | 67.7 | 47.5 | 66.1 | 76.0 |

best accuracy without exceeding D-FINE (small)'s latency. Note that these RF-DETRmodels are different than those reported in Tables 2 and 12.

As shown in Table 11, the DINOv2-S backbone family initially surpasses D-FINE in mAP@50:95 but fails to maintain this advantage at larger model sizes, suggesting that its scaling strategy is less effective than D-FINE's manual design. In contrast, the DINOv2-B backbone family shows the opposite trend, where the performance gap between D-FINE and RF-DETR narrows as latency increases. This implies that at higher latencies, the DINOv2-B based RF-DETR models could surpass D-FINE (and indeed RF-DETR (2x-large) outperforms D-FINE on mAP 50:95). Importantly, expanding the D-FINE model family would require substantial additional engineering effort, whereas extending the RF-DETR model family is straightforward; higher-latency variants can be sampled directly from the same NAS search without re-training. We present the COCO and RF100-VL results of our larger variants in Tables 12, 13, and 14. We also include an RF-DETR Max variant on each dataset to show our method's maximum performance with latency less than 100ms, a scale other model families don't reach.

# F  PER-KNOB SENSITIVITY ANALYSIS

We evaluate the impact of varying resolution and patch size in Figure 5. Both curves follow a clear Pareto frontier, and are consistent with findings from prior work like FlexiViT (Beyer et al., 2023). RF-DETR is able to gracefully interpolate between seen (blue circles) and unseen (red stars) configurations during inference. Importantly, unseen configurations closely track the trend established by the seen configurations, demonstrating that RF-DETR generalizes beyond the model configurations encountered during training.

Table 13: **COCO Segmentation Evaluation for Larger Model Variants.** We present RF-DETR's performance for L, XL, and 2XL sizes on the COCO segmentation benchmark below. We find that scaling up RF-DETR yields considerable performance improvements. In contrast, YOLOv8 and YOLOv11 do not significantly improve with scale.

| Model | Size | # Params. | GFLOPS | Latency (ms) | AP | $AP_{50}$ | $AP_{75}$ | $AP_S$ | $AP_M$ | $AP_L$ |
|---|---|---|---|---|---|---|---|---|---|---|
| **Real-Time Instance Segmentation w/ NMS** | | | | | | | | | | |
| YOLOv8 (Jocher et al., 2023) | L | 46.0M | 220.5 | 9.7 | 39.0 | 60.5 | 41.7 | 18.0 | 44.7 | 57.8 |
| YOLOv11 (Jocher et al., 2024) | L | 27.6M | 132.2 | 8.3 | 39.5 | 61.5 | 42.1 | 18.6 | 45.5 | 59.4 |
| YOLOv8 (Jocher et al., 2023) | XL | 71.8M | 344.1 | 14.0 | 39.5 | 61.3 | 42.1 | 18.9 | 45.6 | 58.8 |
| YOLOv11 (Jocher et al., 2024) | XL | 62.1M | 296.4 | 13.7 | 40.1 | 62.4 | 42.6 | 18.8 | 46.4 | 60.1 |
| **End-to-End Real-Time Instance Segmentation** | | | | | | | | | | |
| RF-DETR (Ours) | L | 36.2M | 151.1 | 8.8 | 47.1 | 70.5 | 50.9 | 28.4 | 52.1 | 65.6 |
| RF-DETR (Ours) | XL | 38.1M | 260.0 | 13.5 | 48.8 | 72.2 | 53.1 | 30.6 | 53.3 | 65.9 |
| RF-DETR (Ours) | 2XL | 38.6M | 435.3 | 21.8 | 49.9 | 73.1 | 54.5 | 33.9 | 54.1 | 65.7 |
| RF-DETR (Ours) | Max | 40.1M | 1668.2 | 95.6 | 50.5 | 74.0 | 55.4 | 34.6 | 54.2 | 65.4 |

Table 14: **RF100-VL Detection Evaluation for Larger Model Variants.** We present RF-DETR's performance for L, XL, and 2XL sizes on RF100-VL below. Notably, RF-DETR (x-large) beats D-FINE by 0.5 AP. Fine-tuning RF-DETR (x-large) improves performance by an additional 0.4 AP.

| Model | Size | # Params. | GFLOPS | Latency (ms) | AP | $AP_{50}$ | $AP_{75}$ | $AP_S$ | $AP_M$ | $AP_L$ |
|---|---|---|---|---|---|---|---|---|---|---|
| **Real-Time Object Detection w/ NMS** | | | | | | | | | | |
| YOLOv8 (Jocher et al., 2023) | L | 43.7M | 165.2 | 7.9 | 56.5 | 82.1 | 61.1 | 7.1 | 46.0 | 48.9 |
| YOLOv11 (Jocher et al., 2024) | L | 25.3M | 86.9 | 6.4 | 56.5 | 82.2 | 61.0 | 6.4 | 45.5 | 49.0 |
| YOLOv8 (Jocher et al., 2023) | XL | 68.2M | 257.8 | 11.2 | 56.5 | 82.3 | 61.0 | 6.6 | 45.7 | 47.9 |
| YOLOv11 (Jocher et al., 2024) | XL | 56.9M | 194.9 | 10.3 | 56.2 | 81.7 | 60.8 | 6.1 | 45.9 | 48.1 |
| **End-to-End Real-Time Object Detection** | | | | | | | | | | |
| RT-DETR (Zhao et al., 2024) | R50 | 42M | 136 | 8.4 | 61.7 | 87.7 | 66.9 | 38.1 | 57.1 | 69.4 |
| LW-DETR (Chen et al., 2024a) | L | 46.8M | 137.5 | 6.8 | 61.5 | 87.4 | 67.0 | 37.1 | 56.4 | 69.0 |
| D-FINE (Peng et al., 2024) | L | 31M | 91 | 7.5 | 61.6 | 86.4 | 67.2 | 37.8 | 56.5 | 70.1 |
| RF-DETR (Ours) | L | 34.1M | 119.1 | 6.2 | 62.0 | 88.1 | 67.3 | 36.9 | 57.1 | 70.2 |
| RF-DETR (Ours) w/ Fine-Tuning | L | 34.1M | 119.1 | 6.2 | 62.3 | 88.2 | 67.4 | 37.1 | 57.2 | 70.3 |
| RT-DETR (Zhao et al., 2024) | R101 | 76M | 259 | 11.9 | 61.0 | 87.4 | 66.2 | 36.6 | 56.3 | 68.2 |
| LW-DETR (Chen et al., 2024a) | XL | 118.0M | 342.5 | 13.0 | 62.1 | 87.9 | 67.6 | 37.4 | 57.1 | 70.2 |
| D-FINE (Peng et al., 2024) | XL | 59.3 | 76.8 | 11.4 | 62.2 | 86.9 | 68.0 | 37.6 | 57.4 | 69.7 |
| RF-DETR (Ours) | XL | 35.0M | 199.0 | 9.7 | 62.6 | 88.5 | 67.9 | 39.0 | 57.8 | 70.4 |
| RF-DETR (Ours) w/ Fine-Tuning | XL | 35.0M | 199.0 | 9.7 | 63.0 | 88.7 | 68.2 | 38.8 | 58.2 | 70.6 |
| RF-DETR (Ours) | 2XL | 123.5M | 410.2 | 15.6 | 63.3 | 88.9 | 69.0 | 38.7 | 58.2 | 71.6 |
| RF-DETR (Ours) w/ Fine-Tuning | 2XL | 123.5M | 410.2 | 15.6 | 63.5 | 89.0 | 69.2 | 38.9 | 58.3 | 71.7 |

## G  IMPACT OF NAS FINE-TUNING ON COCO

We find that fine-tuning after NAS provides limited benefit for COCO. We posit that the NAS "architecture augmentation" acts as a strong regularizer, and additional training without this regularization leads to degraded performance. Specifically, when models are pre-trained with strong regularization, removing the regularization during fine-tuning leads to overfitting. As shown in Tables 15 and 16, this trend is consistent across both detection and segmentation tasks. Interestingly, models trained on RF100-VL benefit more from fine-tuning, likely because they require more than 100 epochs to

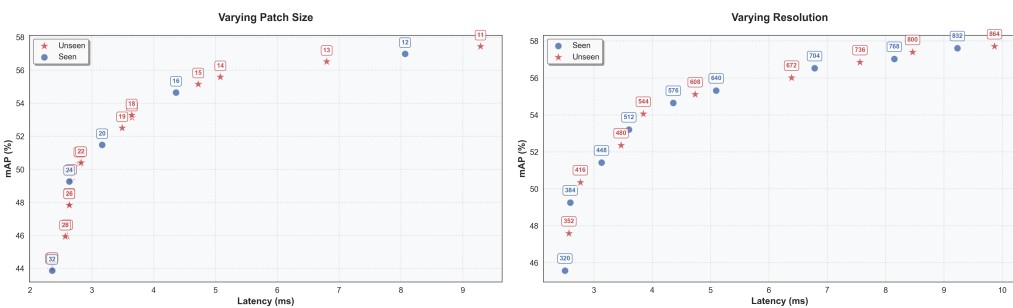

Figure 5: **Per Knob Sensitivity Analysis.** Despite never seeing certain resolutions and patch sizes, RF-DETR is able to gracefully interpolate to novel model configurations.

converge. In such cases, we posit that reducing the total number of NAS configurations during training, or training for more than 100 epochs with weight-sharing NAS can improve performance.

Table 15: **COCO Detection Fine-Tuning Evaluation.** We find that fine-tuning after NAS provides limited benefit for COCO detection, particularly for larger model sizes.

| Model | Size | # Params. | GFLOPS | Latency (ms) | AP | $AP_{50}$ | $AP_{75}$ | $AP_S$ | $AP_M$ | $AP_L$ |
|---|---|---|---|---|---|---|---|---|---|---|
| **End-to-End Real-Time Object Detectors** | | | | | | | | | | |
| RF-DETR (Ours) | N | 30.5M | 31.9 | 2.3 | 48.0 | 67.0 | 51.4 | 25.2 | 53.5 | 70.0 |
| RF-DETR (Ours) w/ Fine-Tuning | N | 30.5M | 31.9 | 2.3 | +0.4 | +0.6 | +0.3 | +0.1 | +0.1 | +1.3 |
| RF-DETR (Ours) | S | 32.1M | 59.8 | 3.5 | 52.9 | 71.9 | 57.0 | 32.0 | 58.3 | 73.0 |
| RF-DETR (Ours) w/ Fine-Tuning | S | 32.1M | 59.8 | 3.5 | +0.1 | +0.2 | +0.2 | -0.2 | +0.2 | +0.1 |
| RF-DETR (Ours) | M | 33.7M | 78.8 | 4.4 | 54.7 | 73.5 | 59.2 | 36.1 | 59.7 | 73.8 |
| RF-DETR (Ours) w/ Fine-Tuning | M | 33.7M | 78.8 | 4.4 | +0.0 | +0.1 | +0.0 | -0.1 | +0.1 | -0.1 |
| RF-DETR (Ours) | L | 33.9M | 125.6 | 6.8 | 56.5 | 75.1 | 61.3 | 39.0 | 61.0 | 73.9 |
| RF-DETR (Ours) w/ Fine-Tuning | L | 33.9M | 125.6 | 6.8 | +0.0 | +0.0 | +0.0 | -0.1 | +0.1 | +0.1 |
| RF-DETR (Ours) | XL | 126.4M | 299.3 | 11.5 | 58.6 | 77.4 | 63.8 | 40.3 | 63.9 | 76.2 |
| RF-DETR (Ours) w/ Fine-Tuning | XL | 126.4M | 299.3 | 11.5 | +0.3 | +0.1 | +0.2 | +0.5 | +0.4 | +0.1 |
| RF-DETR (Ours) | 2XL | 126.9M | 438.4 | 17.2 | 60.1 | 78.5 | 65.5 | 43.2 | 64.9 | 76.2 |
| RF-DETR (Ours) w/ Fine-Tuning | 2XL | 126.9M | 438.4 | 17.2 | +0.1 | +0.0 | +0.3 | +0.5 | +0.2 | +0.1 |

Table 16: **COCO Segmentation Fine-Tuning Evaluation.** We find that fine-tuning after NAS provides limited benefit for COCO segmentation, particularly for larger model sizes.

| Model | Size | # Params. | GFLOPS | Latency (ms) | AP | $AP_{50}$ | $AP_{75}$ | $AP_S$ | $AP_M$ | $AP_L$ |
|---|---|---|---|---|---|---|---|---|---|---|
| **End-to-End Real-Time Object Detectors** | | | | | | | | | | |
| RF-DETR-Seg. (Ours) | N | 33.6M | 50.0 | 3.4 | 40.3 | 63.0 | 42.6 | 16.3 | 45.3 | 63.6 |
| RF-DETR-Seg. w/ Fine-Tuning (Ours) | N | 33.6M | 50.0 | 3.4 | +0.1 | +0.4 | +0.0 | -0.5 | +0.2 | +0.7 |
| RF-DETR-Seg. (Ours) | S | 33.7M | 70.6 | 4.4 | 43.1 | 66.2 | 45.9 | 21.9 | 48.5 | 64.1 |
| RF-DETR w/ Fine-Tuning (Ours) | S | Did | Not | Improve | - | - | - | - | - | - |
| RF-DETR-Seg. (Ours) | M | 35.7M | 102.0 | 5.9 | 45.3 | 68.4 | 48.8 | 25.5 | 50.4 | 65.3 |
| RF-DETR w/ Fine-Tuning (Ours) | M | Did | Not | Improve | - | - | - | - | - | - |
| RF-DETR (Ours) | L | 36.2M | 151.1 | 8.8 | 47.1 | 70.5 | 50.9 | 28.4 | 52.1 | 65.6 |
| RF-DETR (Ours) w/ Fine-Tuning | L | Did | Not | Improve | - | - | - | - | - | - |
| RF-DETR (Ours) | XL | 38.1M | 260.0 | 13.5 | 48.8 | 72.2 | 53.1 | 30.6 | 53.3 | 65.9 |
| RF-DETR (Ours) w/ Fine-Tuning | XL | Did | Not | Improve | - | - | - | - | - | - |
| RF-DETR (Ours) | 2XL | 38.6M | 435.3 | 21.8 | 49.9 | 73.1 | 54.5 | 33.9 | 54.1 | 65.7 |
| RF-DETR (Ours) w/ Fine-Tuning | 2XL | Did | Not | Improve | - | - | - | - | - | - |

# H    IMPACT OF DATASET CHARACTERISTICS ON TUNABLE KNOBS

We evaluate the impact of different dataset characteristics on optimal network configurations for RF-DETR (medium) on RF100-VL in Table 17 and Figure 6. We compare different combinations of object size, number of spatial locations, number of decoder layers, number of windows, number of classes, number of annotations, objects per image, and number of queries below. We do not expect these relationships to be linear, but expect that they will be monotonic (e.g. non-zero slope). For example, we find strong correlations between the number of classes and number of decoder layers, objects per image and number of queries, spatial locations and number of windows. Notably, we do not find strong correlations between object size and number of decoder layers, number of annotations and number of decoder layers, and objects per image and number of decoder layers

Table 17: **Regression Analysis.** We evaluate the linear relationships between various dataset characteristics and model parameters. Although these correlations are non-linear, the line-of-best-fit helps explain general trends. P-values indicate the significance of the correlation.

| Relationship | Slope | Intercept | R-squared | P-value | Std. Error |
|---|---|---|---|---|---|
| Average Object Size vs Number of Spatial Locations | -0.009 | 0.384 | 0.148 | 0.000 | 0.002 |
| Average Object Size vs Number of Decoder Layers | -0.000 | 0.044 | 0.000 | 0.971 | 0.006 |
| Average Object Size vs Number of Windows | -0.008 | 0.065 | 0.019 | 0.170 | 0.006 |
| Number of Classes vs Number of Decoder Layers | 0.837 | 2.125 | 0.026 | 0.106 | 0.513 |
| Number of Annotations vs Number of Decoder Layers | 698.763 | 6654.795 | 0.001 | 0.706 | 1848.733 |
| Average Objects per Image vs Number Decoder Layers | 0.163 | 7.618 | 0.000 | 0.857 | 0.904 |
| Average Objects per Image vs Number of Queries | 0.020 | 4.457 | 0.019 | 0.171 | 0.015 |
| Number of Spatial Locations vs Number of Windows | 1.722 | 32.984 | 0.492 | 0.000 | 0.177 |

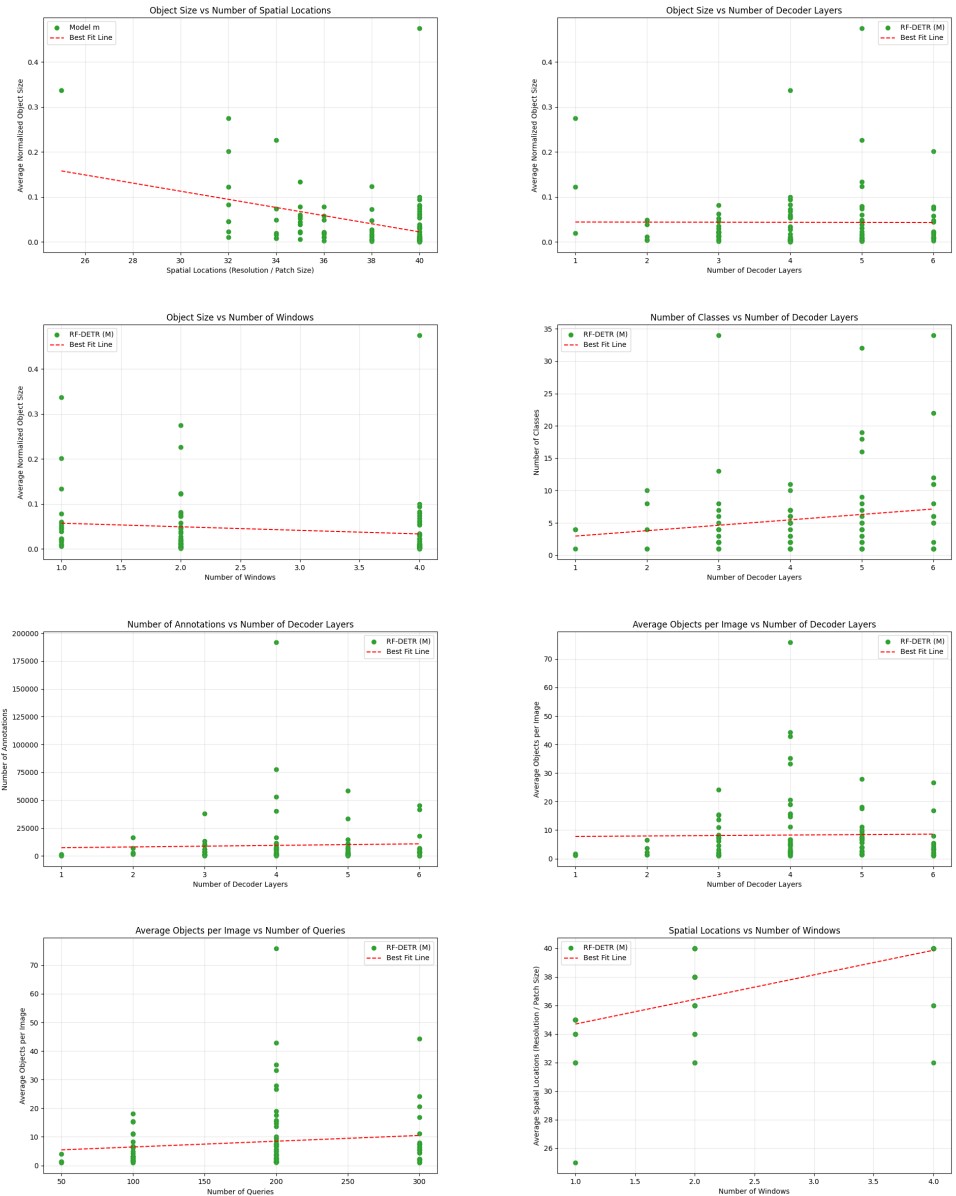

Figure 6: **Impact of Dataset Characteristics on Tunable Knobs**. We visualize the correlation between key dataset characteristics and several tunable architectural knobs above. Across all subplots, individual points represent different datasets within RF100-VL while dashed lines show corresponding linear trends. Overall, the results indicate that object-centric properties of datasets such as average object size, number of classes, object density, and total annotations tend to have only modest influence on architectural choices like the number of decoder layers, number of windows, number of queries, and spatial resolution. Slight positive or negative trends appear in some cases (e.g., more classes or more objects per image loosely correlating with deeper decoders or higher query counts), but the scatter remains wide, suggesting no strong deterministic relationship. These findings highlight that while dataset characteristics offer some intuition for selecting model hyperparameters, optimal configurations ultimately depend on a combination of factors rather than any single dataset attribute.

## I   IMPACT OF FIXED ARCHITECTURE ON RF100-VL

We evaluate the impact of transferring a NAS architecture optimized for COCO to RF100-VL in Table 18 and Figure 7. We find that these fixed architecture models perform remarkably well without further dataset-specific NAS. Specifically, RF-DETR (large) model with a fixed architecture achieves the best performance among all prior real-time models on COCO. However, dataset-specific NAS yields significant improvements. Notably, the performance delta from LW-DETR to the fixed architecture is comparable to the improvement from the fixed architecture to the NAS-optimized model on the target dataset for nano, small, and medium scale models.

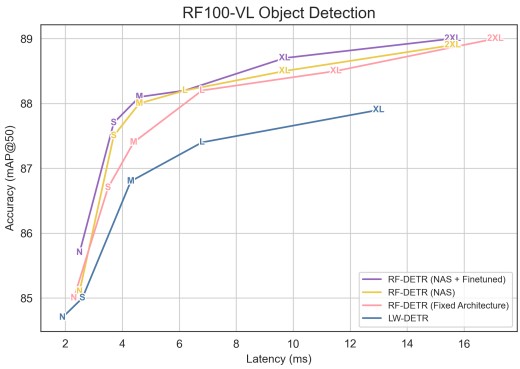

Figure 7: **Ablating Fixed Architecture RF100-VL**. We evaluate the benefit of dataset-specific NAS by transferring the COCO-optimized RF-DETR architecture to RF100-VL. Although the fixed architecture was not tuned for RF100-VL, it still outperforms LW-DETR. Running NAS directly on RF100-VL further improves performance over the fixed architecture. Additional fine-tuning provides consistent gains across all model sizes, with particularly strong improvements for smaller models. This is consistent with our observations on COCO object detection.

Table 18: **RF100-VL Fixed Architecture Evaluation.** We evaluate the transfer of architectures optimized for COCO to RF100-VL. Fixed architecture models perform well without additional dataset-specific NAS, with the RF-DETR (large) model achieving the best performance among prior real-time models. However, dataset-specific NAS provides significant further gains.

| Model | Size | # Params. | GFLOPS | Latency (ms) | AP | $AP_{50}$ | $AP_{75}$ | $AP_S$ | $AP_M$ | $AP_L$ |
|---|---|---|---|---|---|---|---|---|---|---|
| **End-to-End Real-Time Object Detectors** | | | | | | | | | | |
| RF-DETR (Ours) Fixed Architecture | N | 30.5M | 31.9 | 2.3 | 57.7 | 85.0 | 61.9 | 30.8 | 51.5 | 67.4 |
| RF-DETR (Ours) | N | 31.2M | 34.5 | 2.5 | 57.8 | 85.1 | 62.5 | 30.1 | 52.2 | 67.2 |
| RF-DETR w/ Fine-Tuning (Ours) | N | 31.2M | 34.5 | 2.5 | 58.6 | 85.7 | 63.0 | 31.0 | 53.2 | 67.6 |
| RF-DETR (Ours) Fixed Architecture | S | 32.1M | 59.8 | 3.5 | 60.2 | 86.7 | 65.0 | 34.2 | 54.4 | 68.9 |
| RF-DETR (Ours) | S | 33.5M | 62.4 | 3.7 | 60.9 | 87.5 | 66.1 | 34.2 | 55.7 | 69.6 |
| RF-DETR w/ Fine-Tuning (Ours) | S | 33.5M | 62.4 | 3.7 | 61.2 | 87.7 | 66.1 | 34.9 | 55.6 | 69.5 |
| RF-DETR (Ours) Fixed Architecture | M | 33.7M | 78.8 | 4.4 | 61.2 | 87.4 | 66.4 | 35.8 | 56.1 | 69.8 |
| RF-DETR (Ours) | M | 33.6M | 91.0 | 4.6 | 61.7 | 88.0 | 66.9 | 35.8 | 56.5 | 70.0 |
| RF-DETR w/ Fine-Tuning (Ours) | M | 33.6M | 91.0 | 4.6 | 62.0 | 88.1 | 67.1 | 36.2 | 56.4 | 70.2 |
| RF-DETR (Ours) w/ Fixed Architecture | L | 33.9M | 125.6 | 6.8 | 62.2 | 88.2 | 67.8 | 37.7 | 57.0 | 70.5 |
| RF-DETR (Ours) | L | 34.1M | 119.1 | 6.2 | 62.0 | 88.1 | 67.3 | 36.9 | 57.1 | 70.2 |
| RF-DETR (Ours) w/ Fine-Tuning | L | 34.1M | 119.1 | 6.2 | 62.3 | 88.2 | 67.4 | 37.1 | 57.2 | 70.3 |
| RF-DETR (Ours, DINOv2-Base) w/ Fixed Architecture | XL | 126.4M | 299.3 | 11.5 | 62.9 | 88.5 | 68.6 | 37.0 | 57.5 | 71.3 |
| RF-DETR (Ours) | XL | 35.0M | 199.0 | 9.7 | 62.6 | 88.5 | 67.9 | 39.0 | 57.8 | 70.4 |
| RF-DETR (Ours) w/ Fine-Tuning | XL | 35.0M | 199.0 | 9.7 | 63.0 | 88.7 | 68.2 | 38.8 | 58.2 | 70.6 |
| RF-DETR (Ours, DINOv2-Base) w/ Fixed Architecture | 2XL | 126.9M | 438.4 | 17.1 | 63.2 | 89.0 | 69.3 | 38.4 | 58.4 | 71.5 |
| RF-DETR (Ours, DINOv2-Base) | 2XL | 123.5M | 410.2 | 15.6 | 63.3 | 88.9 | 69.0 | 38.7 | 58.2 | 71.6 |
| RF-DETR (Ours, DINOv2-Base) w/ Fine-Tuning | 2XL | 123.5M | 410.2 | 15.6 | 63.5 | 89.0 | 69.2 | 38.9 | 58.3 | 71.7 |

## J   ABLATION ON BACKBONE ARCHITECTURE WITH RF20-VL

In Table 19, we reproduce our ablation on the impact of backbone architecture on downstream model performance (Table 6) on RF20-VL. All trends from the main paper hold.

Table 19: **Ablation on Backbone RF20-VL.**

| LW-DETR (M) + Gentler Hyperparameters | # Params. | GFLOPS | Latency (ms) | AP | $AP_{50}$ | $AP_{75}$ | $AP_S$ | $AP_M$ | $AP_L$ |
|---|---|---|---|---|---|---|---|---|---|
| w/ CAEv2 ViT/S-16-Truncated Backbone | 28.3M | 83.7 | 4.3 | 64.4 | 92.2 | 71.2 | 33.8 | 56.8 | 71.9 |
| w/ DINOv2 ViT/S-14 Backbone | 32.3M | 78.2 | 4.6 | 65.2 | 92.5 | 72.8 | 37.8 | 58.2 | 72.6 |
| w/ SigLIPv2 ViT/B-32 Backbone* | 105.1M | 81.6 | 4.5 | 62.2 | 91.0 | 67.8 | 28.9 | 54.1 | 70.3 |
| w/ SAM2 Hiera-S Backbone* | 44.0M | 109.1 | 11.2 | 65.2 | 92.5 | 72.7 | 37.8 | 58.2 | 72.1 |

## K  ANALYSIS ON BUFFERING

We further analyze the impact of buffering on the relative ordering of inference speed in Table 20. Notably, we find that buffering beyond 200ms does not change latency measurements. However, we acknowledge that adding a 200ms buffer after every forward pass considerably increases overall inference time. Future work should consider alternatives to buffering to address power throttling.

Table 20: **Analysis on Buffering.** We evaluate models with different amounts of buffering between consecutive forward passes. We find that buffering beyond 200ms does not provide any additional stability to latency measurements.

| Model | mAP | 0 ms | 200 ms | 400 ms | 800 ms |
|---|---|---|---|---|---|
| YOLOv8 (M) | 47.3 | 5.5 ms | 5.4 ms | 5.4 ms | 5.4 ms |
| YOLOv11 (M) | 48.4 | 5.0 ms | 5.0 ms | 5.1 ms | 5.1 ms |
| RT-DETR (R18) | 49.0 | 4.4 ms | 4.4 ms | 4.4 ms | 4.4 ms |
| LW-DETR (M) | 52.6 | 4.5 ms | 4.3 ms | 4.3 ms | 4.3 ms |
| D-FINE (M) | 54.9 | 5.7 ms | 5.4 ms | 5.4 ms | 5.4 ms |
| RF-DETR (M) | 54.7 | 4.7 ms | 4.4 ms | 4.4 ms | 4.4 ms |

## L  DISCUSSION ON NOTABLE DISCOVERED ARCHITECTURES

Several trends emerge from our weight-sharing NAS. First, we note that all tunable "knobs" are used when defining Pareto-optimal model families, validating our search space. This suggests that expanding the search space may further improve downstream performance.

Across Pareto-optimal models, patch size is consistent within model families. For example, the optimal patch size for RF-DETRs with a DINOv2-S backbone is 16, RF-DETRs with a DINOv2-B backbone is 20, and RF-DETR-Segs with a DINOv2-S backbone is 12. Pareto-optimal models also jointly scale encoder and decoder compute: patch size, number of windows, and resolution impact the encoder, while decoder depth, and number of queries affect the decoder. For RF-DETR-Seg, scaling resolution impacts the segmentation head. We find that using 2 windows in the encoder is typically optimal and resolution scales within a model family as we increase latency. On COCO, RF-DETR scales decoder depth while keeping the number of queries fixed, while RF-DETR-Seg simultaneously scales both axes. This likely reflects a minimum viable segmentation head depth; to offset its latency, RF-DETR-Seg reduces its total number of queries, yielding a thin, deep decoder, in contrast to RF-DETRs wide, shallow decoder.

Next, we find that RF-DETR's performance is more correlated with the total number of spatial locations (e.g. resolution divided by patch size) rather than resolution or patch size alone. Scaling resolution with a fixed patch size yields similar results to scaling patch size with a fixed resolution, since vision transformers are agnostic to absolute input resolution after the patchify-and-project operation. To verify this, we constructed an alternative model family with fixed resolution (640) and varied patch sizes to preserve the total number of spatial locations. Specifically, we evaluate RF-DETR (nano) with a patch size of 27, RF-DETR (small) with a patch size of 21, and RF-DETR (medium) with a patch size of 18. Surprisingly, all model results are nearly identical to the Pareto-optimal family. Notably, patch sizes of 27 and 18 were unseen during training, demonstrating RF-DETR's strong generalization to novel patch sizes (Beyer et al., 2023). However, we find that this trend does not hold for RF-DETR-Seg because segmentation features are always upsampled to $\frac{1}{4}$ of the input image resolution. As a result, scaling RF-DETR-Seg's input resolution affects both the number of spatial locations and the segmentation feature resolution. Specifically, RF-DETR-Seg (nano, small, medium) uses input resolutions of 312, 384, and 432 with patch size 12, yielding segmentation feature resolutions of 78, 96, and 108 and 26, 32, and 36 spatial locations, respectively. In contrast, increasing patch size alone (e.g., patch size 16 at input resolution 576) preserves spatial locations while increasing segmentation feature resolution. As a result, although RF-DETR (medium) and RF-DETR-Seg (medium) both use 36 spatial locations, RF-DETR-Seg operates at lower input resolution, demonstrating that coupling segmentation feature resolution with input resolution shifts the Pareto-optimal operating point.

Further, we find that most Pareto-optimal RF-DETR models perform best with 2 windows, whereas LW-DETR achieves the best performance with 4 windows. We attribute this difference to how each architecture handles class tokens. LW-DETRs CAEv2 backbone omits the class token, while RF-

DETR's DINOv2 backbone relies on it as a key part of pre-training. To make windowed attention compatible with class tokens, we duplicate the class token for each window. During global attention, window-level class tokens attend to one another, while all other tokens attend to all class tokens. In practice, RF-DETR (nano), RF-DETR (small), and RF-DETR (medium) all use 2 windows, since duplicating class tokens for additional windows reduces runtime efficiency. As a result, unlike LW-DETR, RF-DETR does not benefit from scaling to 4 windows.

Lastly, we note that dataset characteristics influence optimal discovered architectures. We find that the optimal low latency models on RF100-VL tend to use fewer queries than the COCO models of equivalent latency. We attribute this to RF100-VL datasets having fewer objects per image than COCO.

## M  VISUALIZING MODEL PREDICTIONS

We visualize model predictions from RF-DETR (nano) and relevant detection and segmentation baselines in Figure 8. We find that RF-DETR (nano) predicts fewer false positives (e.g. mistaking `sign post` for `person`). Similarly, RF-DETR-Seg. (nano) predicts more precise object boundaries.

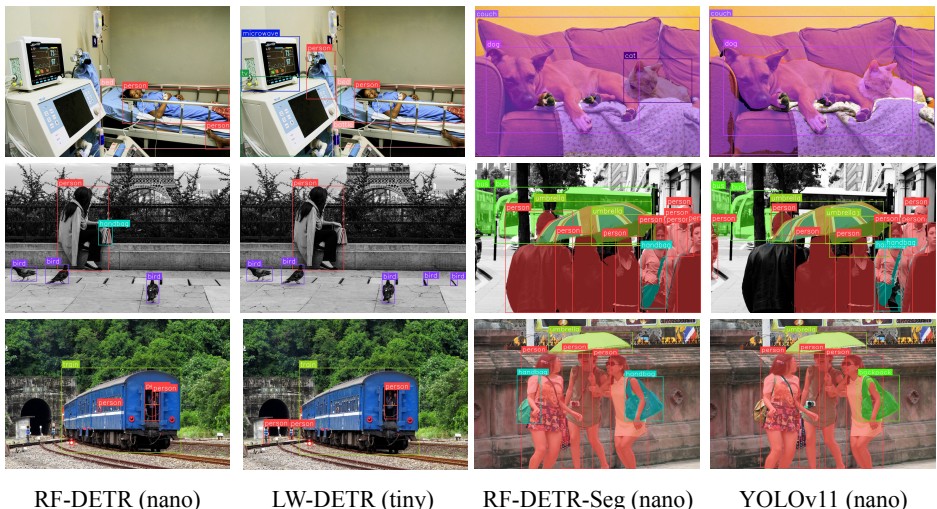

| RF-DETR (nano) | LW-DETR (tiny) | RF-DETR-Seg (nano) | YOLOv11 (nano) |

Figure 8: **Visualizing Model Predictions**. On the left, we compare detections from RF-DETR (nano) and LW-DETR (tiny). On the right, we compare instance segmentation masks from RF-DETR-Seg (nano) and YOLOv11 (nano)