# OpenReview forum: "RF-DETR: Neural Architecture Search for Real-Time Detection Transformers"
_ICLR.cc/2026/Conference — ICLR 2026 Poster_

### Official Review · Reviewer_HtQ3 · 2025-10-31

**Soundness:** 3
**Presentation:** 2
**Contribution:** 2
**Rating:** 4
**Confidence:** 5

**Summary:**

This paper introduces RF-DETR, a lightweight specialist object detector that uses NAS to discover accuracy-latency Pareto curves for target datasets. The key contributions are: (1) a family of NAS-based detection and segmentation models that outperform prior real-time methods on COCO and Roboflow100-VL; (2) exploration of "tunable knobs" in weight-sharing NAS for end-to-end detection, improving transferability; and (3) a standardized latency evaluation protocol to address reproducibility issues.

**Strengths:**

+ The novelty of this work is acceptable. It applys end-to-end weight-sharing NAS to DETR-based detectors, which has been underexplored. Unlike prior NAS methods focused on image classification or backbones, RF-DETR optimizes full detection pipelines, including segmentation heads.
+ The paper includes extensive experiments on COCO and RF100-VL. Ablations validate design choices, such as backbone replacements and NAS components. Results show consistent improvements.
+ The authors have clarified their motivations. Some figures can help understanding, and the language is precise. Some method details have been provided.
+ This method outperforms some famous baselines like YOLO and D-FINE, and the NAS framework allows customization for diverse hardware.

**Weaknesses:**

- The generalization beyond COCO and RF100-VL cannot be confirmed.​​ The experiments focus on two benchmarks, but the claim of generalizability to "any target dataset" is not fully validated. Testing on diverse domains would demonstrate broader applicability. The paper notes that hyperparameters may overfit to COCO-like data and more cross-dataset results would alleviate this concern.
- The paper lacks a theoretical analysis of why weight-sharing NAS generalizes well to unseen architectures. For example, it does not provide robustness analysis for the NAS mechanism.
- The authors may further figure out the specific gaps in existing NAS methods for object detection in the introduction. While it mentions overfitting to COCO, it does not thoroughly explain why current NAS approaches fail in detection tasks or how RF-DETR uniquely addresses these issues, especially NAS is not a new concept or tool.

**Questions:**

- The weight-sharing NAS involves sampling configurations during training. What is the total training time compared to a non-NAS baseline?
- RF-DETR is positioned as a specialist detector, but how does it compare to fine-tuned VLMs in terms of accuracy-latency trade-offs?
- The buffering method reduces throttling, but did the authors consider other techniques? Please explain why buffering was preferred over alternatives.
- The authors may consider the weakness above and address these key concerns.

---

> ### Author Response · Authors · 2025-11-23
> **Response to Reviewer HtQ3**
>
> Thank you for your review, we address your concerns below.
>
> **Claims of Generalization to “Any Target Dataset” and Cross-Dataset Results**
>
> We evaluate RF-DETR on RF100-VL, a collection of _100 diverse detection datasets_, as a proxy for performance on “any target dataset”. RF100-VL is sourced “from Roboflow Universe, a community-driven platform that hosts diverse open-source datasets created to solve real-world computer vision tasks. With more than 500, 000 public datasets spanning medical imaging, agriculture, robotics, and manufacturing, we focus on selecting high-quality datasets not commonly found in internet-scale pre-training (e.g. COCO, Objects365, GoldG, CC4M) to better assess VLM generalization to rare concepts. When selecting candidates for RF100-VL, we prioritized datasets where images contained multiple objects, ensuring more realistic evaluation beyond classification.” [1].  Notably, SAM3 [2] also evaluates on RF100-VL to make similar claims about generalization to diverse target datasets (SAM3’s Appendix F.3).
>
> [1] Roboflow100-VL: A Multi-Domain Object Detection Benchmark for Vision-Language Models. Robicheaux and Popov et al. NeurIPS D&B 2025.
>
> [2] SAM 3: Segment Anything with Concepts. SAM3 Team.
>
> **Analysis on NAS Generalization to Unseen Architecture**
>
> Great suggestion! We evaluate the impact of varying resolution and patch size below. Notably, despite never seeing certain resolutions and patch sizes, RF-DETR is able to gracefully interpolate between these configurations.
>
> _Varying Resolution_
> | **Resolution** | **mAP**   | **Latency (ms)** | **Unseen** |
> |------------|-------|---------------|--------|
> | 320        | 45.57 | 2.50          | No     |
> | 352        | 47.58 | 2.57          | Yes    |
> | 384        | 49.25 | 2.60          | No     |
> | 416        | 50.34 | 2.77          | Yes    |
> | 448        | 51.41 | 3.13          | No     |
> | 480        | 52.35 | 3.47          | Yes    |
> | 512        | 53.21 | 3.60          | No     |
> | 544        | 54.05 | 3.85          | Yes    |
> | 576        | 54.64 | 4.36          | No     |
> | 608        | 55.11 | 4.74          | Yes    |
> | 640        | 55.31 | 5.10          | No     |
> | 672        | 56.00 | 6.39          | Yes    |
> | 704        | 56.52 | 6.78          | No     |
> | 736        | 56.84 | 7.56          | Yes    |
> | 768        | 57.02 | 8.15          | No     |
> | 800        | 57.40 | 8.46          | Yes    |
> | 832        | 57.61 | 9.23          | No     |
> | 864        | 57.70 | 9.87          | Yes    |
>
> _Varying Patch Size_
> | **Patch Size** | **mAP**   | **Latency** | **Unseen** |
> |------------|-------|---------|--------|
> | 11         | 57.44 | 9.29    | Yes    |
> | 12         | 56.98 | 8.07    | No     |
> | 13         | 56.51 | 6.80    | Yes    |
> | 14         | 55.58 | 5.08    | Yes    |
> | 15         | 55.15 | 4.72    | Yes    |
> | 16         | 54.64 | 4.36    | No     |
> | 17         | 53.14 | 3.65    | Yes    |
> | 18         | 53.27 | 3.64    | Yes    |
> | 19         | 52.50 | 3.49    | Yes    |
> | 20         | 51.47 | 3.17    | No     |
> | 21         | 50.35 | 2.80    | Yes    |
> | 22         | 50.39 | 2.83    | Yes    |
> | 23         | 49.31 | 2.66    | Yes    |
> | 24         | 49.25 | 2.63    | No     |
> | 25         | 47.86 | 2.63    | Yes    |
> | 26         | 47.83 | 2.63    | Yes    |
> | 27         | 45.96 | 2.60    | Yes    |
> | 28         | 45.93 | 2.57    | Yes    |
> | 29         | 43.91 | 2.36    | Yes    |
> | 30         | 43.93 | 2.34    | Yes    |
> | 31         | 43.97 | 2.36    | Yes    |
> | 32         | 43.86 | 2.36    | No     |
>
> **Clarifying RF-DETR’s Novelty**
>
> To the best of our knowledge, RF-DETR is the first weight-sharing NAS to use pre-trained foundational backbones. Unlike OFA, our approach does not require training schedulers, allowing us to effectively transfer to diverse target datasets without extensive hyperparameter search. Moreover, OFA also uses architecture-level schedulers to modify the architecture during training, which does not allow for using pre-trained foundational backbones with fixed architectures. Further, although “NAS has been previously explored in the context of image classification and for model sub-components like detector backbones and FPNs” (L51), RF-DETR explores end-to-end weight-sharing NAS for object detection and segmentation.
>
> **Details on NAS Training**
>
> The total training time is roughly 2 to 4 times as long as a non-NAS baseline, depending on the target dataset. However, RF-DETR can generate all size configurations from this single training run, while other non-NAS baselines must be re-trained for each new model size.

---

> > ### Author Response · Authors · 2025-11-23
> >
> > **Comparing Accuracy-Latency Tradeoff of RF-DETR vs. Fine-Tuned VLM**
> >
> > We compare RF-DETR’s performance compared to fine-tuned VLMs in Table 4, and include additional results on SAM3 below. Notably, RF-DETR (M) achieves comparable performance to GroundingDINO, LLMDet and SAM3 on RF100-VL at a fraction of the run-time. Note that SAM3 reports their latency on a H200 GPU (which has roughly 18x faster FP16 performance [1]), while latencies for GroundingDINO, LLMDet, and RF-DETR are reported with a T4 GPU. We expect that scaling RF-DETR to larger model sizes will improve performance further.
> >
> > | **Model**           | **Latency (ms)** | **mAP**  |
> > |-----------------|--------------|------|
> > | Grounding DINO  | 309.9        | 62.3 |
> > | LLMDet          | 308.4        | 62.3 |
> > | SAM3            | 30*          | 61.6 |
> > | RF-DETR (M)     | 4.7          | 61.5 |
> >
> > [1] https://tinyurl.com/t4-h200-comparison
> >
> > **Analysis on Buffering**
> >
> > We chose to address GPU power throttling with buffering because it was simple to implement and sufficiently addressed the issue. As Reviewer fnAm highlights, our goal is to introduce “practical” benchmarking protocols that "practitioners can actually follow”. We further analyze the impact of buffering on the relative ordering inference speed below. Notably, we find that buffering beyond 200ms does not change latency measurements. However, we acknowledge that adding a 200ms buffer after every forward pass considerably increases overall inference time. Future work should consider alternatives to buffering to address power throttling.
> >
> > | Model         | 0 ms                     | 200 ms                   | 400 ms                   | 800 ms                   |
> > |---------------|--------------------------|--------------------------|--------------------------|--------------------------|
> > | YOLOv8 (M)    | mAP: 47.28 Latency: 5.53 | mAP: 47.28 Latency: 5.38 | mAP: 47.28 Latency: 5.40 | mAP: 47.28 Latency: 5.41 |
> > | YOLOv11 (M)   | mAP: 48.37 Latency: 5.00 | mAP: 48.37 Latency: 5.04 | mAP: 48.37 Latency: 5.06 | mAP: 48.37 Latency: 5.06 |
> > | RT-DETR (R18) | mAP: 49.02 Latency: 4.44 | mAP: 49.02 Latency: 4.35 | mAP: 49.02 Latency: 4.36 | mAP: 49.02 Latency: 4.36 |
> > | LW-DETR (M)   | mAP: 52.55 Latency: 4.46 | mAP: 52.55 Latency: 4.32 | mAP: 52.55 Latency: 4.30 | mAP: 52.55 Latency: 4.31 |
> > | D-FINE (M)    | mAP: 54.94 Latency: 5.67 | mAP: 54.94 Latency: 5.37 | mAP: 54.94 Latency: 5.38 | mAP: 54.94 Latency: 5.38 |
> > | RF-DETR (M)   | mAP: 54.74 Latency: 4.74 | mAP: 54.74 Latency: 4.40 | mAP: 54.74 Latency: 4.40 | mAP: 54.74 Latency: 4.40 |

---

> ### Comment · Reviewer_HtQ3 · 2025-11-26
>
> Thanks for the authors response. The new analysis is interesting and useful, and their clarification is also acceptable. I will raise my score after discussing with other reviewers. It would be better if the authors include the analysis in their manuscript.

---

### Official Review · Reviewer_FU3r · 2025-10-31

**Soundness:** 3
**Presentation:** 4
**Contribution:** 2
**Rating:** 4
**Confidence:** 5

**Summary:**

The paper introduces RF-DETR, a fast, closed-vocabulary DETR that uses weight-sharing NAS to pick the best accuracy–latency setup (tuning resolution, patch size, decoder depth, queries, and windowing) after a single fine-tune—no extra retraining needed. It swaps in a DINOv2 ViT backbone, adds a lightweight mask head (RF-DETR-Seg), and standardizes latency benchmarking (buffering, same FP model, and counting NMS/mask conversion). Results on COCO and Roboflow100-VL show a new real-time Pareto frontier—e.g., the nano model beats D-FINE (nano) by about 5 AP and the medium model is near GroundingDINO (tiny) while roughly 60× faster—suggesting many recent detectors are implicitly over-optimized for COCO.

**Strengths:**

Originality here is pragmatic: the paper smartly fuses weight-sharing NAS with DETR and a foundation-model backbone, plus a no-nonsense latency protocol—less theory, more removal of real deployment roadblocks. Quality is strong for an engineering paper: careful ablations, honest discussion of TensorRT variance and FP16 pitfalls, and fair apples-to-apples latency (counting NMS/mask steps) bolster credibility. Clarity is good—the “knobs” are concrete, the scheduler-free recipe is easy to replicate, and the deployment story (post-hoc grid search, decoder/query truncation) is clean. Significance is high for practice: it moves the real-time Pareto frontier, cuts the cost of retargeting to new hardware/domains, and exposes COCO-centric benchmarking biases; the results on RF100-VL make the transfer claims believable.

**Weaknesses:**

1) Gains appear driven mainly by stronger pretraining (DINOv2, O365+SAM2) and a stable recipe, making NAS’s unique contribution unclear.
2) Size taxonomy is misleading: the “nano” model (26.9M params) is far larger than baseline “nano” (3–4M), confounding size vs latency comparisons.
3) The latency protocol (200ms buffering; TensorRT/FlashAttention) stabilizes single-shot timing but can bias rankings and does not reflect sustained throughput.
4) YOLO baselines on RF100-VL are disadvantaged by COCO-tuned thresholds and multi-/single-class NMS mismatches, likely underestimating their performance.
5) Mixed-precision/export inconsistencies (FP32 accuracy vs FP16 latency; modified ONNX export) undermine strict parity across methods.

**Questions:**

1) Size taxonomy and fairness: Can you clarify the naming (e.g., “nano”) and provide comparisons under fixed latency/parameter/FLOP budgets or a plot at fixed latency caps?
2) Latency and throughput: Can you report sustained throughput (QPS) under continuous load without the 200 ms buffer, include full pipeline timing (pre/post-processing, NMS/mask conversion), and show sensitivity to buffer length and FlashAttention usage?
3) YOLO baselines on RF100-VL: Did you use multi-class NMS and dataset-specific threshold tuning consistent with original inference? Can you provide threshold/post-processing sensitivity analysis to verify robustness?
4) Mixed-precision/export parity: Are all methods evaluated with identical artifacts (same precision, same export path)? Can you disclose the modified export scripts (e.g., ONNX opset 17) and quantify FP16 accuracy changes?
5) NAS cost and coverage: What are the GPU-hours for training/search, how much of the search space is sampled, and how do “unseen” subnets perform statistically?
6) Method details: How is “encoder confidence” defined for query dropping, and what criteria or default policy govern decoder truncation at inference?

---

> ### Author Response · Authors · 2025-11-23
> **Response to Reviewer FU3r**
>
> Thank you for your review, we address your concerns below.
>
> **Evaluating the Contribution of Backbone vs. NAS**
>
> We agree that a majority of RF-DETR’s improvements on COCO are driven by the DINOv2 backbone and Objects-365 pre-training. However, NAS allows RF-DETR to generalize to diverse target datasets in RF100-VL without per-dataset hyperparameter tuning. Different from prior real-time detectors that must re-train from scratch for each new size configuration, RF-DETR is able to generate new model configurations by sampling different operating points along the pareto-optimal curve from a single training run (cf. Figure 1). Lastly, prior work significantly modifies their backbone between model sizes, making it difficult to repurpose existing fixed-size foundational backbones. In contrast, NAS allows RF-DETR to adapt fixed-size foundational backbones for arbitrary operating points along the Pareto frontier.
>
> **Clarification on Sizing Taxonomy**
>
> As you correctly point out, RF-DETR (N) has 26.9M parameters and a run-time of 2.3 ms, while YOLOv8 (N) has 3.2M parameters and a run-time of 2.3 ms. This suggests that parameter count is a poor proxy of run-time. Therefore, we follow LW-DETR’s protocol and group model sizes based on latency. Notably, RF-DETR’s parameter count does not scale as fast as YOLOv8’s parameter count due to our weight-sharing NAS approach, further motivating our latency-based grouping. We plot RF-DETR’s continuous accuracy-latency curve in Figure 1, highlighting its superior accuracy at fixed latency.
>
> **Upper Bound on YOLO Baselines**
>
> Great suggestion! We do not evaluate YOLO baselines with multi-class NMS  since the first-party implementation only supports it for validation (not inference). Furthermore, multi-class NMS is not supported in ONNX or TensorRT, making it difficult to accurately measure latency. Therefore, we compare RF-DETR with YOLOv8 and YOLOv11’s upper bound performance without considering latency (as reported in RF100-VL [1]) using FP32 weights and multi-class NMS consistent with the original implementation below. Notably, YOLOv8 and YOLOv11’s performance with multi-class NMS is still lower than RF-DETR’s performance on RF100-VL.
>
> | **Method**        | **AP**   |
> |---------------|------|
> | YOLOv8 (N)    | 55.4 |
> | YOLOv11 (N)   | 56.1 |
> | RF-DETR (N)   | 57.3 |
> | YOLOv8 (S)    | 56.5 |
> | YOLOv11 (S)   | 57.0 |
> | RF-DETR (S)   | 60.6 |
> | YOLOv8 (M)    | 56.9 |
> | YOLOv11 (M)   | 57.0 |
> | RF-DETR (M)   | 61.5 |
>
> [1] Roboflow100-VL: A Multi-Domain Object Detection Benchmark for Vision-Language Models. Robicheaux and Popov et al. NeurIPS D&B 2025.
>
> **Number of Candidates Evaluated during Search**
>
> RF-DETR’s total training time is roughly 2 to 4 times as long as a non-NAS baseline, depending on the target dataset. However, RF-DETR can generate all size configurations from this single training run, while other non-NAS baselines must be re-trained for each new model size. We evaluate 6,468 network configurations (11 resolutions * 7 patch sizes * 7 decoder layers * 3 windows * 4 query settings) during architecture search. We estimate that search time uses approximately 10,000 GPU Hours (200 GPUs * 48 hours).

---

> > ### Author Response · Authors · 2025-11-23
> >
> > **Analysis on NAS Generalization to Unseen Architecture**
> >
> > Great suggestion! We evaluate the impact of varying resolution and patch size below. Notably, despite never seeing certain resolutions and patch sizes, RF-DETR is able to gracefully interpolate between these configurations.
> >
> > _Varying Resolution_
> > | **Resolution** | **mAP**   | **Latency (ms)** | **Unseen** |
> > |------------|-------|---------------|--------|
> > | 320        | 45.57 | 2.50          | No     |
> > | 352        | 47.58 | 2.57          | Yes    |
> > | 384        | 49.25 | 2.60          | No     |
> > | 416        | 50.34 | 2.77          | Yes    |
> > | 448        | 51.41 | 3.13          | No     |
> > | 480        | 52.35 | 3.47          | Yes    |
> > | 512        | 53.21 | 3.60          | No     |
> > | 544        | 54.05 | 3.85          | Yes    |
> > | 576        | 54.64 | 4.36          | No     |
> > | 608        | 55.11 | 4.74          | Yes    |
> > | 640        | 55.31 | 5.10          | No     |
> > | 672        | 56.00 | 6.39          | Yes    |
> > | 704        | 56.52 | 6.78          | No     |
> > | 736        | 56.84 | 7.56          | Yes    |
> > | 768        | 57.02 | 8.15          | No     |
> > | 800        | 57.40 | 8.46          | Yes    |
> > | 832        | 57.61 | 9.23          | No     |
> > | 864        | 57.70 | 9.87          | Yes    |
> >
> > _Varying Patch Size_
> > | **Patch Size** | **mAP**   | **Latency** | **Unseen** |
> > |------------|-------|---------|--------|
> > | 11         | 57.44 | 9.29    | Yes    |
> > | 12         | 56.98 | 8.07    | No     |
> > | 13         | 56.51 | 6.80    | Yes    |
> > | 14         | 55.58 | 5.08    | Yes    |
> > | 15         | 55.15 | 4.72    | Yes    |
> > | 16         | 54.64 | 4.36    | No     |
> > | 17         | 53.14 | 3.65    | Yes    |
> > | 18         | 53.27 | 3.64    | Yes    |
> > | 19         | 52.50 | 3.49    | Yes    |
> > | 20         | 51.47 | 3.17    | No     |
> > | 21         | 50.35 | 2.80    | Yes    |
> > | 22         | 50.39 | 2.83    | Yes    |
> > | 23         | 49.31 | 2.66    | Yes    |
> > | 24         | 49.25 | 2.63    | No     |
> > | 25         | 47.86 | 2.63    | Yes    |
> > | 26         | 47.83 | 2.63    | Yes    |
> > | 27         | 45.96 | 2.60    | Yes    |
> > | 28         | 45.93 | 2.57    | Yes    |
> > | 29         | 43.91 | 2.36    | Yes    |
> > | 30         | 43.93 | 2.34    | Yes    |
> > | 31         | 43.97 | 2.36    | Yes    |
> > | 32         | 43.86 | 2.36    | No     |
> >
> > **Additional Details on Decoder Layers and Queries**
> >
> > Queries are ranked by the sigmoid of the class logits at the output of the encoder. We perform grid search over all architecture combinations (shown below) and select the pareto-optimal configurations based on latency and mAP. We will clarify this in our updated draft.
> >
> > _Inference Configurations_
> > - Resolutions: 320, 384, 448, 512, 576, 640, 704, 768, 832, 896, 960
> > - Patch Sizes: 8, 10, 12, 16, 20, 24, 32
> > - Number of Decoder Layers: 0, 1, 2, 3, 4, 5, 6
> > - Number of Windows: 1, 2, 4
> > - Number of Queries: 50, 100, 200, 300

---

> > > ### Comment · Reviewer_FU3r · 2025-11-26
> > >
> > > The rebuttal answered most of my questions, and the extra analyses were helpful. Some issues remain only partly addressed, but they are not critical. I think this is a solid and substantial engineering effort with detailed experiments that clearly support the ideas. I would also emphasize that incorporating the promised clarifications and analyses into the revised paper is important, as several details currently only appear in the rebuttal. I will keep my original score.

---

### Official Review · Reviewer_c7Lr · 2025-11-01

**Soundness:** 4
**Presentation:** 4
**Contribution:** 3
**Rating:** 8
**Confidence:** 4

**Summary:**

The paper proposes a weight-sharing neural architecture search space for finding realtime DETR-style object detectors that are pareto-optimal w.r.t. accuracy and runtime. The resulting models achieve state-of-the-art real-time results on the COCO and Roboflow100-VL object detection benchmarks.

**Strengths:**

The paper is well written and approachable. The search space is elegantly designed and evidently yields state-of-the-art results. The paper makes a set of great points about suboptimal benchmarking practices in prior work and makes an effort to perform fair comparisons.

**Weaknesses:**

While the paper does provide significant contributions, it could support more ablations and experiments. For example, I would have loved to see a thorough discussion of how much each of the “tunable knobs” contributes to favorable accuracy-runtime tradeoffs, and how much pareto-optimal “knob-settings” vary between datasets. Furthermore, it would be interesting to see whether specific dataset characteristics, like the prevalence of small objects, have an impact on knobs like patch size. I furthermore find it highly interesting that DINOv2 performs much better on “small datasets”, but there are no systematic comparisons between other backbones.

**Questions:**

* Instance segmentation:
  * How is the proposed approach different from Mask DINO?
  * “Our segmentation head bilinearly interpolates the output of the FPN and learns a lightweight projector to generate a pixel embedding map” \- Where is the FPN coming from? I thought this was using a DINO backbone?
* There are two inconsistent definitions of latencies at which RF-DETR outperforms prior work: “for all latencies” and “all latencies ≤ 40 ms”. Which of these is correct?

---

> ### Author Response · Authors · 2025-11-23
> **Response to Reviewer c7Lr**
>
> Thank you for your review, we address your concerns below.
>
> **Per-Knob Sensitivity Analysis**
>
> Great suggestion! We evaluate the impact of varying resolution and patch size below. Notably, despite never seeing certain resolutions and patch sizes, RF-DETR is able to gracefully interpolate between these configurations.
>
> _Varying Resolution_
> | **Resolution** | **mAP**   | **Latency (ms)** | **Unseen** |
> |------------|-------|---------------|--------|
> | 320        | 45.57 | 2.50          | No     |
> | 352        | 47.58 | 2.57          | Yes    |
> | 384        | 49.25 | 2.60          | No     |
> | 416        | 50.34 | 2.77          | Yes    |
> | 448        | 51.41 | 3.13          | No     |
> | 480        | 52.35 | 3.47          | Yes    |
> | 512        | 53.21 | 3.60          | No     |
> | 544        | 54.05 | 3.85          | Yes    |
> | 576        | 54.64 | 4.36          | No     |
> | 608        | 55.11 | 4.74          | Yes    |
> | 640        | 55.31 | 5.10          | No     |
> | 672        | 56.00 | 6.39          | Yes    |
> | 704        | 56.52 | 6.78          | No     |
> | 736        | 56.84 | 7.56          | Yes    |
> | 768        | 57.02 | 8.15          | No     |
> | 800        | 57.40 | 8.46          | Yes    |
> | 832        | 57.61 | 9.23          | No     |
> | 864        | 57.70 | 9.87          | Yes    |
>
> _Varying Patch Size_
> | **Patch Size** | **mAP**   | **Latency** | **Unseen** |
> |------------|-------|---------|--------|
> | 11         | 57.44 | 9.29    | Yes    |
> | 12         | 56.98 | 8.07    | No     |
> | 13         | 56.51 | 6.80    | Yes    |
> | 14         | 55.58 | 5.08    | Yes    |
> | 15         | 55.15 | 4.72    | Yes    |
> | 16         | 54.64 | 4.36    | No     |
> | 17         | 53.14 | 3.65    | Yes    |
> | 18         | 53.27 | 3.64    | Yes    |
> | 19         | 52.50 | 3.49    | Yes    |
> | 20         | 51.47 | 3.17    | No     |
> | 21         | 50.35 | 2.80    | Yes    |
> | 22         | 50.39 | 2.83    | Yes    |
> | 23         | 49.31 | 2.66    | Yes    |
> | 24         | 49.25 | 2.63    | No     |
> | 25         | 47.86 | 2.63    | Yes    |
> | 26         | 47.83 | 2.63    | Yes    |
> | 27         | 45.96 | 2.60    | Yes    |
> | 28         | 45.93 | 2.57    | Yes    |
> | 29         | 43.91 | 2.36    | Yes    |
> | 30         | 43.93 | 2.34    | Yes    |
> | 31         | 43.97 | 2.36    | Yes    |
> | 32         | 43.86 | 2.36    | No     |
>
> **Impact of Dataset Characteristics on Knobs**
>
> We evaluate the impact of different dataset characteristics on optimal network configurations for RF100-VL. We compare different combinations of object size, number of spatial locations, number of decoder layers, number of windows, number of classes, number of annotations, objects per image, and number of queries below. We do not expect these relationships to be linear, but expect that they will be monotonic (e.g. non-zero slope).  For example, we find strong correlations between the number of classes and number of decoder layers, objects per image and number of queries, spatial locations and number of windows. Notably, we do not find strong correlations between object size and number of decoder layers, number of annotations and number of decoder layers, and objects per image and number of decoder layers. We will include plots of these relationships in the updated draft.
>
> | **Relationship**                              | **Slope**    | **Intercept**  | **R-Squared** | **P-Value** | **STD. Error** |
> |-------------------------------------------|----------|------------|-----------|---------|------------|
> | Object Size vs. # Spatial Locs.           | -0.009   | 0.384      | 0.148     | 0.000   | 0.002      |
> | Object Size vs. # Decoder Layer           | -0.000   | 0.044      | 0.000     | 0.971   | 0.006      |
> | Object Size vs. # Windows                 | -0.008   | 0.065      | 0.019     | 0.170   | 0.006      |
> | # Classes vs. # Decoder Layers            | 0.837    | 2.125      | 0.026     | 0.106   | 0.513      |
> | # Annotations vs. # Decoder Layers        | 698.763  | 6654.795   | 0.001     | 0.706   | 1848.733   |
> | Objects per Image vs. # Decoder Layers    | 0.163    | 7.618      | 0.000     | 0.857   | 0.904      |
> | Objects Per Image vs. Num Queries         | 0.020    | 4.457      | 0.019     | 0.171   | 0.015      |
> | Spatial Locations vs. # Windows           | 1.722    | 32.984     | 0.492     | 0.000   | 0.177      |
>
> **Additional Details on Instance Segmentation Head**
>
> The FPN is the projector block at the output of the DINOv2 backbone (cf. Figure 2). Unlike MaskDINO, we do not incorporate multi-scale backbone features into our segmentation head to minimize latency. In addition, our segmentation head can drop decoder layers to trade off accuracy and latency. We will clarify this in our updated draft.
>
> **Minor Clarifications About Performance Claims**
>
> RF-DETR outperforms prior work for all latencies less than 40ms. We will clarify this in our updated draft.

---

> > ### Comment · Reviewer_c7Lr · 2025-11-25
> >
> > The authors rebuttal addressed all of my concerns (per-knob sensitivity analysis) and questions (model architecture and performance claims).

---

### Official Review · Reviewer_fnAm · 2025-11-01

**Soundness:** 4
**Presentation:** 4
**Contribution:** 3
**Rating:** 6
**Confidence:** 3

**Summary:**

The paper proposes a weight-sharing, once-for-all style search over DETR variants (“RF-DETR”) to obtain real-time operating points along an accuracy–latency Pareto curve. A single base model is trained while exposing a compact set of architectural knobs (input resolution, patch size, decoder depth, number of queries, windowed vs. global attention). After training, sub-networks are selected by validation without per-subnet fine-tuning. Experiments on COCO and a broader multi-dataset suite show consistent gains over strong real-time baselines, and an instance-segmentation extension indicates broader utility.

**Strengths:**

1. A practical blend of weight sharing, once for all selection, and DETR knob tuning that feels deployable, with latency protocols that practitioners can actually follow.
2. Clear empirical gains against strong real time baselines on COCO and on a broader evaluation suite, and the instance segmentation extension indicates the idea transfers beyond detection.
3. A search space whose knobs are easy to grasp, with figures and ablations that make the design choices legible.
4. A simple path to high quality low latency detectors without fine tuning of each subnet, plus useful guidance on latency measurement such as throttling mitigation and artifact consistency.

**Weaknesses:**

1. The way subnets are sampled during training and the policy or grid used for post training selection are not specified clearly, which hurts reproducibility and interpretation.
2. The paper does not make clear when or how decoder layers and queries are dropped during training, whether losses are reweighted across depths, or how queries are ranked at test time.
3. The contribution reads as incremental relative to once for all and weight sharing NAS in vision, and a tighter comparison to prior NAS for detection and backbones is needed to clarify what is new beyond engineering.
4. Some comparisons mix backbones and pretraining regimes, so stronger parity baselines or controlled reruns would better isolate the benefit of the proposed recipe.
5. It is unclear how a fixed mined subnet transfers to unseen datasets or domains without reselection, and a cross dataset test would strengthen the robustness claim.
6. Per knob sensitivity and stability evidence are limited, for example queries versus AP at fixed FLOPs and the interaction of resolution and patch size, and reporting variance with error bars or minimum median maximum would help.

**Questions:**

1. How are sub-networks sampled during training (uniform over knobs, FLOPs-aware, or constrained)? Any coupling constraints to avoid pathological combinations?
2. Are layers/queries randomly dropped during training to mimic inference-time truncation? Is there loss re-weighting across depths? How exactly are queries ranked at test time?
3. How many candidates are evaluated during selection, what is the wall-clock/energy budget, and is the same operating point reused across datasets/hardware or re-selected each time?
4. Could you add controlled re-runs (or a table) where backbones and pre-training are aligned across methods to isolate the effect of your approach?
5. Could you report results where a single subnet chosen on COCO is evaluated unchanged on other datasets to assess transfer.
6. Could you provide per-knob sensitivity plots and report variance across random seeds and multiple TensorRT engine builds?
7. Will you release export scripts, calibration settings, the buffering/throttling harness, and the exact list of selected sub-networks?

---

> ### Author Response · Authors · 2025-11-23
> **Response to Reviewer fnAm**
>
> Thank you for your review, we address your concerns below.
>
> **Additional Details on Sampling Subnets**
>
> We sample all subnets uniformly during training. Although pathological combinations like high resolution and small patch size can spike GPU memory usage, we do not explicitly address this issue. We include the explicit sampling grid used during training and inference below. We will clarify this in our updated draft.
>
> _Training Configurations_
> - Image Resolutions: 320, 384, 448, 512, 576, 640, 704, 768, 832, 896, 960
> - Patch Sizes: 8, 10, 12, 16, 20, 24, 32
> - Number of Decoder Layers: 6
> - Number of Windows: 1, 2, 4
> - Number of Query: 300
>
> _Inference Configurations_
> - Image Resolutions: 320, 384, 448, 512, 576, 640, 704, 768, 832, 896, 960
> - Patch Sizes: 8, 10, 12 ,14, 16, 20, 24, 32
> - Number of Decoder Layers: 0, 1, 2, 3, 4, 5, 6
> - Number of Windows: 1, 2, 4
> - Number of Queries: 50, 100, 200, 300
>
> **Additional Details on Decoder Layers and Queries**
>
> We train all decoder layers with standard detector losses during training and only drop layers at inference. Therefore, we do not need to re-weight losses across decoder layers. Similarly, we use all queries during training and only drop queries at inference. Queries are ranked by the sigmoid of the class logits at the output of the encoder. We will clarify this in our updated draft.
>
> **Number of Candidates Evaluated during Search**
>
> We evaluate 6,468 network configurations (11 resolutions * 7 patch sizes * 7 decoder layers * 3 windows * 4 query settings) during architecture search. We estimate that NAS uses approximately 700kWH (70 Watts per T4 GPU * 200 GPUs * 48 hours).  Notably, although RF-DETR models have inference speeds on the order of a few milliseconds, our search time is considerably increased because we buffer for 200ms after every forward pass to accurately estimate latency. We re-sample the pareto-optimal operating point for each dataset.
>
> **Aligning Pre-Training Across Baselines**
>
> All transformer-based baselines are pre-trained on Objects365. Therefore, our method’s improvements can be primarily attributed to our DINOv2 backbone and weight-sharing NAS.
>
> **Clarifying RF-DETR’s Novelty**
>
> To the best of our knowledge, RF-DETR is the first weight-sharing NAS to use pre-trained foundational backbones. Unlike OFA, our approach does not require training schedulers, allowing us to effectively transfer to diverse target datasets without extensive hyperparameter search. Moreover, OFA also uses architecture-level schedulers to modify the architecture during training, which does not allow for using pre-trained foundational backbones with fixed architectures. Further, although “NAS has been previously explored in the context of image classification and for model sub-components like detector backbones and FPNs” (L51), RF-DETR explores end-to-end weight-sharing NAS for object detection and segmentation.
>
> **Evaluating the Transferability of Fixed Subnets to Unseen Datasets**
>
> Great suggestion! We evaluate the transferability of dataset-specific architectures by directly training the COCO-optimized RF-DETR architecture for all datasets in RF100-VL without additional NAS. Although the fixed architecture was not tuned for RF100-VL, it still outperforms LW-DETR. Running NAS directly on RF100-VL further improves performance over the fixed architecture, highlighting the benefits of our approach.
>
> | **Model**                         | **Latency** | **mAP**  |
> |-------------------------------|---------|------|
> | LW-DETR (T)                   | 1.9     | 57.1 |
> | RF-DETR (N) w/ COCO NAS       | 2.3     | 57.7 |
> | RF-DETR (N) w/ RF100-VL NAS   | 2.5     | 57.6 |
> | LW-DETR (S)                   | 2.6     | 57.4 |
> | RF-DETR (S) w/ COCO NAS       | 3.5     | 60.2 |
> | RF-DETR (S) w/ RF100-VL       | 3.7     | 60.7 |
> | LW-DETR (M)                   | 4.3     | 59.6 |
> | RF-DETR (M) w/ COCO NAS       | 4.4     | 61.2 |
> | RF-DETR (M) w/ RF100-VL       | 4.6     | 61.5 |

---

> > ### Author Response · Authors · 2025-11-23
> >
> > **Per-Knob Sensitivity Analysis**
> >
> > Great suggestion! We evaluate the impact of varying resolution and patch size below. Notably, despite never seeing certain resolutions and patch sizes, RF-DETR is able to gracefully interpolate between these configurations.
> >
> > _Varying Resolution_
> > | **Resolution** | **mAP**   | **Latency (ms)** | **Unseen** |
> > |------------|-------|---------------|--------|
> > | 320        | 45.57 | 2.50          | No     |
> > | 352        | 47.58 | 2.57          | Yes    |
> > | 384        | 49.25 | 2.60          | No     |
> > | 416        | 50.34 | 2.77          | Yes    |
> > | 448        | 51.41 | 3.13          | No     |
> > | 480        | 52.35 | 3.47          | Yes    |
> > | 512        | 53.21 | 3.60          | No     |
> > | 544        | 54.05 | 3.85          | Yes    |
> > | 576        | 54.64 | 4.36          | No     |
> > | 608        | 55.11 | 4.74          | Yes    |
> > | 640        | 55.31 | 5.10          | No     |
> > | 672        | 56.00 | 6.39          | Yes    |
> > | 704        | 56.52 | 6.78          | No     |
> > | 736        | 56.84 | 7.56          | Yes    |
> > | 768        | 57.02 | 8.15          | No     |
> > | 800        | 57.40 | 8.46          | Yes    |
> > | 832        | 57.61 | 9.23          | No     |
> > | 864        | 57.70 | 9.87          | Yes    |
> >
> > _Varying Patch Size_
> > | **Patch Size** | **mAP**   | **Latency** | **Unseen** |
> > |------------|-------|---------|--------|
> > | 11         | 57.44 | 9.29    | Yes    |
> > | 12         | 56.98 | 8.07    | No     |
> > | 13         | 56.51 | 6.80    | Yes    |
> > | 14         | 55.58 | 5.08    | Yes    |
> > | 15         | 55.15 | 4.72    | Yes    |
> > | 16         | 54.64 | 4.36    | No     |
> > | 17         | 53.14 | 3.65    | Yes    |
> > | 18         | 53.27 | 3.64    | Yes    |
> > | 19         | 52.50 | 3.49    | Yes    |
> > | 20         | 51.47 | 3.17    | No     |
> > | 21         | 50.35 | 2.80    | Yes    |
> > | 22         | 50.39 | 2.83    | Yes    |
> > | 23         | 49.31 | 2.66    | Yes    |
> > | 24         | 49.25 | 2.63    | No     |
> > | 25         | 47.86 | 2.63    | Yes    |
> > | 26         | 47.83 | 2.63    | Yes    |
> > | 27         | 45.96 | 2.60    | Yes    |
> > | 28         | 45.93 | 2.57    | Yes    |
> > | 29         | 43.91 | 2.36    | Yes    |
> > | 30         | 43.93 | 2.34    | Yes    |
> > | 31         | 43.97 | 2.36    | Yes    |
> > | 32         | 43.86 | 2.36    | No     |
> >
> > **Releasing Code**
> >
> > Yes, we will release all export scripts, calibration settings, and buffering harnesses. We provide the pareto-optimal model configurations on COCO below.
> >
> > | **Model**              | **Resolution** | **Patch Size** | **Windows** | **Decoder Layers** | **Queries** |
> > |--------------------|------------|------------|---------|----------------|---------|
> > | RF-DETR (N)        | 384        | 16         | 2       | 2              | 300     |
> > | RF-DETR (S)        | 512        | 16         | 2       | 3              | 300     |
> > | RF-DETR (M)        | 576        | 16         | 2       | 4              | 300     |
> > | RF-DETR-Seg (N)    | 312        | 12         | 2       | 4              | 200     |
> > | RF-DETR-Seg (S)    | 384        | 12         | 2       | 4              | 200     |
> > | RF-DETR-Seg (M)    | 432        | 12         | 2       | 4              | 200     |

---

### Author Response · Authors · 2025-11-27
**Summary of Reviews**

We are thrilled reviewers agree that RF-DETR “moves the real-time Pareto frontier, cuts the cost of retargeting to new domains and exposes COCO-centric benchmarking biases” (FU3r). Reviewers also appreciate that our work is “practical” (fnAm), “well written and approachable” (c7Lr), with “extensive experiments on COCO and RF100-VL” (HtQ3). We thank all reviewers for their feedback as it has materially improved our paper. We summarize key reviewer concerns and our responses below.

**Major Reviewer Concerns**
- _Clarifying RF-DETR’s Novelty_. Reviewers FnAm and HtQ3 ask us to clarify RF-DETR’s novelty compared to prior real-time detectors and NAS-based methods. To the best of our knowledge, RF-DETR is the first weight-sharing NAS to use pre-trained foundational backbones for real-time object detection. Unlike OFA [1], our approach does not require training schedulers, allowing us to effectively transfer to diverse target datasets without extensive hyperparameter search. Moreover, OFA also uses architecture-level schedulers to modify the architecture during training, which does not allow for using pre-trained foundational backbones with fixed architectures. Further, although “NAS has been previously explored in the context of image classification and for model sub-components like detector backbones and FPNs” (L51), RF-DETR explores end-to-end weight-sharing NAS for object detection and segmentation.
- _Contribution of NAS_. Reviewer Fu3r correctly points out that a majority of RF-DETR’s performance gains on COCO are driven by the DINOv2 backbone and Objects-365 pre-training and asks about the contribution of NAS in our method. We demonstrate that NAS allows RF-DETR to generalize to diverse target datasets in RF100-VL without per-dataset hyperparameter tuning (Table 4). Different from prior real-time detectors that must re-train from scratch for each new size configuration, RF-DETR is able to generate new model configurations by sampling different operating points along the pareto-optimal curve from a _single training run_ (Figure 1). Lastly, prior work significantly modifies their backbone between model sizes, making it difficult to repurpose existing fixed-size foundational backbones. In contrast, NAS allows RF-DETR to adapt fixed-size foundational backbones for arbitrary operating points along the Pareto frontier.
- _Generalization to “Any Target Dataset”_. Reviewer HtQ3 asks us to justify our claim that RF-DETR generalizes to “any target dataset”. We evaluate RF-DETR on Roboflow100-VL [2], a collection of 100 diverse detection datasets, as a proxy for performance on “any target dataset”.  Notably, SAM3 [3] also evaluates on Roboflow100-VL to make similar claims about generalization to diverse target datasets (SAM3’s Appendix F.3).
- _Additional Ablations_. Reviewers ask for additional experiments on the impact of transferring subnets optimized for COCO to novel datasets (fnAm), a sensitivity analysis of the NAS searchable space (fnAm, c7Lr, Fu3r, HtQ3), an analysis of dataset characteristics on optimal NAS parameters (c7Lr), and an analysis on the impact of buffering on latency measurements (HtQ3). We provide detailed results for all ablations in the reviewer responses below.

[1] Once-for-All: Train One Network and Specialize it for Efficient Deployment. Cai et. al. ICLR 2020

[2] Roboflow100-VL: A Multi-Domain Object Detection Benchmark for Vision-Language Models. Robicheaux and Popov et al. NeurIPS D&B 2025.

[3] SAM 3: Segment Anything with Concepts. SAM3 Team. ArXiv 2025.

**Updates to the Paper**
- Additional Details on Instance Segmentation Head (L208)
- Additional Details on Sampling Subnets During Training: (L247)
- Additional Details on Dropping Decoder Layers: (L259)
- Additional Details on Dropping Queries: (L265)
- Clarification on Sizing Taxonomy (L312)
- Number of Candidates Evaluated During NAS: Appendix A
- Pareto-Optimal Model Configurations for COCO: Appendix A
- Per Knob Sensitivity Analysis: Appendix F
- Impact of Dataset Characteristics on Tunable Knobs: Appendix H
- Evaluating the Transferability of Fixed Subnets to Unseen Datasets: Appendix I
- Analysis on Buffering: Appendix K

Reviewers acknowledge that we have addressed their major concerns. We provide a revised draft of our paper above, which includes the suggested changes.

---

### Meta-Review · Area_Chair_MxyA · 2025-12-31

**Summary:**

The authors use weight-sharing neural architecture search to train a compute-optimal detection transformer. They show impressive results in terms of compute-accuracy trade-offs on both COCO and the challenging and new Roboflow 100 dataset.

Although the results are impressive, reviewers were concerned that there was not sufficient insight into why the method works (ie why is weight-sharing NAS effective here, and what are the downsides of previous NAS methods). Reviewers also requested a number of additional experiments which the authors answered convincingly in the rebuttal.

Overall, the decision is to accept the paper. Please incorporate the rebuttal responses into the camera ready.

The AC also acknowledges that some of the reviews appear to be LLM-generated, and took it into account. And thank you for responding to these reviews in good faith.

**Reviewer Concerns:**

All concerns of Reviewer c7Lr addressed,

Reviewer HtQ3 mostly addressed, except for theoretical analysis / insight of the method.

**Reviewer Scores:**

c7Lr -- already positive. Unlikely to change

HtQ3 -- Increased rating to accept.

FU3r and FnAm appear to be LLM-generated. Ignored

---

### Decision · Program_Chairs · 2026-01-26

Accept (Poster)